# AdaRankGrad: Adaptive Gradient Rank and Moments for Memory-Efficient LLMs Training and Fine-Tuning

**Yehonathan Refael**[1]     **Jonathan Svirsky**[2]     **Boris Shustin**[3]

**Wasim Huleihel**[1]     **Ofir Lindenbaum**[2]

[1]Tel Aviv University     [2]Bar-Ilan University     [3]University of Oxford

## Abstract

Training and fine-tuning large language models (LLMs) come with challenges related to memory and computational requirements due to the increasing size of the model weights and the optimizer states. Various techniques have been developed to tackle these challenges, such as low-rank adaptation (LoRA), which involves introducing a parallel trainable low-rank matrix to the fixed pre-trained weights at each layer. However, these methods often fall short compared to the full-rank weight training approach, as they restrict the parameter search to a low-rank subspace. This limitation can disrupt training dynamics and require a full-rank warm start to mitigate the impact. In this paper, we introduce a new method inspired by a phenomenon we formally prove: as training progresses, the rank of the estimated layer gradients gradually decreases, and asymptotically approaches rank one. Leveraging this, our approach involves adaptively reducing the rank of the gradients during Adam optimization steps, using an efficient online-updating low-rank projections rule. We further present a randomized SVD scheme for efficiently finding the projection matrix. Our technique enables full-parameter fine-tuning with adaptive low-rank gradient updates, significantly reducing overall memory requirements during training compared to state-of-the-art methods while improving model performance in both pretraining and fine-tuning. Finally, we provide a convergence analysis of our method and demonstrate its merits for training and fine-tuning language and biological foundation models. The code is available on GitHub.

## 1    Introduction

Large language models (LLMs) have gained significant attention due to their impressive ability to handle various tasks, such as dialogue-based systems and text completion. Both supervised fine-tuning and additional pre-training can further enhance their performance across tasks and domains. However, training these models presents significant computational and memory challenges. This is because performing the gradient updates requires storing billions of LLM's trainable parameters along with the optimizer state (e.g., gradients and moments). In Adam, for example, the gradients and the estimated first and second moments triple the size of the model itself (Xu et al., 2024; Brown et al., 2022; Kim et al., 2023).

To tackle the challenges associated with LLM fine-tuning, researchers have developed various optimization techniques to reduce memory usage during model training. One key approach that has emerged is Parameter-efficient fine-tuning (PEFT) (Han et al., 2024), which enables the adaptation of pre-trained language models (PLMs) to different tasks without the need to fine-tune all model parameters. A prominent method within PEFT is the Low-Rank Adaptation (LoRA) algorithm, introduced by Hu et al. (2021). LoRA reparameterizes a weight matrix $\mathbf{W} \in \mathbb{R}^{m \times n}$ into $\mathbf{W} = \mathbf{W}_0 + \mathbf{BA}$, where $\mathbf{W}_0$ is a frozen full-rank matrix, and $\mathbf{B} \in \mathbb{R}^{m \times r}$ and $\mathbf{A} \in \mathbb{R}^{r \times n}$ are low-rank adaptors. Since $r \ll \min(m, n)$, the low-rank adaptors $\mathbf{A}$ and $\mathbf{B}$ require fewer trainable parameters, reducing memory usage. LoRA has been widely adopted for fine-tuning, with several

variants emerging, including Adaptive LoRA, which adapts the rank of the matrices during training (Wang et al., 2023), LoRA+, which uses different learning rates for the two matrices (Chen et al., 2023), and Sparse LoRA, which introduces sparsity to the matrices to further reduce computational cost (Xu et al., 2023). These methods have been demonstrated to enhance the efficiency and performance of LLM fine-tuning for various tasks.

Despite its advantages, recent research has identified some limitations of low-rank reparameterization. For example, LoRA may not achieve the same performance levels as full-rank fine-tuning (Meng et al., 2024) and might require initial full-rank model training as a warm-up before effectively utilizing the low-rank subspace (Lialin et al., 2023b). These issues may stem from the fact that optimal weight matrices are not inherently low-rank or from changes in gradient training dynamics introduced by the reparameterization. In addition, LoRA keeps the tuning layer shapes in the base model static without dynamic adjustments. Another approach by He et al. (2022) dynamically adjusts tuning parameters during training, and (Zhang et al., 2023; Svirsky et al., 2024) gradually reduces tuning parameters.

Until recently, the pre-training of large language models (LLMs) has been primarily limited to corporations and governments with substantial computational and memory resources. The significant challenges posed by the enormous memory and computational requirements made it impractical for the average home user. To illustrate this challenge, we take the following as an example. Training a Mistral 7B model from scratch poses substantial memory challenges. Given its 7 billion parameters, a single update step requires approximately 70 GB of memory: 14 GB for the model parameters, 42 GB for Adam optimizer states and gradients, and 14 GB for activations. Consequently, consumer-level GPUs like the NVIDIA RTX 3090, which has 24 GB of VRAM, are inadequate for handling such a large-scale training task.

To overcome this challenge, the study in (Zhao et al., 2024a) introduced a training strategy called GaLore that enables full-parameter learning while being more memory-efficient than traditional low-rank adaptation methods such as LoRA. The core idea behind GaLore is to exploit the slowly changing low-rank structure of the gradient $\mathbf{G} \in \mathbb{R}^{n \times m}$ of the weight matrix $\mathbf{W}$, rather than approximating the weight matrix itself as low-rank. GaLore significantly improved memory efficiency, reducing optimizer state memory usage by up to 65.5%. The following noticeable variant is Q-GaLore (Dettmers et al., 2023), which combines low-rank gradient projection with INT4 quantization to further reduce memory usage, and an additional parallel variant would be ReLoRA (Lialin et al., 2023b) employed in pre-training-by periodically updating $\mathbf{W}_0$ using previously learned low-rank adaptors. We found that Galore to be suboptimal since it arbitrarily requires pre-defining a fixed low-rank size for the gradient projection/low-rank-approximation, while gradients rank gradually diminish during training down to rank one. Additionally, GaLore uses a fixed window size for the number of iterations between updates to the subspace onto which the gradients are projected, keeping this window size constant. Finally, Galore does not transform (adjust) the first and second moments at any update of projection subspace, which we empirically found degrading the potential performance. We suggest an inner transformation scheme of the moments at any projection updates.

**Our approach and theoretical results.** In this paper, we introduce a new training method aimed at optimizing memory efficiency in the training or fine-tuning of large language models (LLMs) while also improving convergence rates and overall performance. Our method leverages two key properties of LLMs. First, we present a novel theoretical finding that shows how the approximate rank of the LLM gradient matrices decreases progressively throughout the training process (even under basic SGD settings), asymptotically approaching rank one. Note that previous studies have only demonstrated an implicit upper bound on the rank of the gradient, which is far from being tight. For example, Zhao et al. (2024a) showed that the rank of the gradient satisfies $\text{rank}(\mathbf{G}^{n \times m}) < \min\{n, m\}/2$. Second, as highlighted in previous research (Gromov et al., 2024; Refael et al., 2024; Jaiswal et al., 2024a), the depth of a layer (how far from input/output) and its architectural design contribute differently to the model's performance. Specifically, when perturbations from the same distribution are applied across various layers, the impact on accuracy varies significantly. This indicates that the optimization steps have less influence on the model's performance for certain layers, depending on their depth and architecture type. Noise in these layers has a smaller impact on the overall task, meaning that the gradients in these layers carry less important information. This results in naturally lower-rank update steps (gradients).

Building upon these two insights and to address the limitations of the LoRA variants, we propose a method that enables full-parameter learning while dramatically reducing memory requirements and computational complexity through adaptive low-rank gradient projections during the Adam update step. For each gradient tensor $\mathbf{G}_t^j \in \mathbb{R}^{r \times n}$ at layer $j \in [L]$ and iteration $t$, AdaRankGrad efficiently identifies a unique set of significant projection directions (subspace) $\mathbf{P}_t^j \in \mathbb{R}^{r_j \times n}$ along which the gradient $\mathbf{G}_t^j$ exhibits the largest changes, where $r_t^j$ is the lowest possible rank to still maintain a predefined information fraction (given threshold) relative to the original, non-projected gradient. Practically, $\mathbf{P}_t^j \mathbf{G}_t^j$ is the low-projected-gradient and $\mathbf{P}_t^{j\top} \mathbf{P}_t^j \mathbf{G}_t^j$ is the best low-rank approximation the of the gradients $\mathbf{G}_t^j$ that embodies the required fraction of its information. The projections $\mathbf{P}_t^j$ are being adaptively updated throughout training (based on the convergence criteria of the gradients on the projected subspace), where their rank $r_t^j$ is dictated by preserving the given information threshold. The method ensures: (1) The method determines the optimal projection dimension for each layer's gradient tensor independently, adjusting dynamically throughout training. This rank adjustment leverages property we prove that the effective dimensionality of the full gradients gradually decreases over time, allowing updates to be performed in a lower-dimensional projection space, thereby reducing memory usage. (2) The projection matrix for each layer's gradients is updated based on a convergence criterion within their respective subspace. This ensures updates occur precisely when needed, avoiding premature or delayed transitions between subspaces and resulting in faster overall convergence.

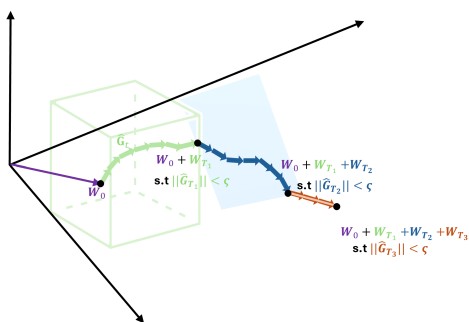

Figure 1: The illustration shows how AdaRankGard 3 is trained. First, the gradients $\mathbf{G}_t$ are projected into a 3D space (in this example), represented as $\hat{\mathbf{G}}_t^{3 \times m} = \mathbf{P}_t^{3 \times n} \mathbf{G}_t^{n \times m}$. As convergence occurs, the gradient's dimension decreases to a 2D space and then to a 1D space. This dimensionality reduction indicates convergence while efficiently using memory.

Table 1: Comparison between AdaRankGrad, GaLore, and LoRA. Assume $\mathbf{W} \in \mathbb{R}^{n \times m}(n \geq m)$, constant rank $r$, and adaptive-rank $r_{\text{adap}}$ (with initial-rank $r_{\text{init}} = r$).

|  | AdaRankGrad | GaLore | LoRA |
|---|---|---|---|
| Weights | $nm$ | $nm$ | $nm + nr + mr$ |
| Optim States ($r_{\text{adap}} < r$) | $nr_{\text{adap}} + 2mr_{\text{adap}}$ | $nr + 2mr$ | $2nr + 2mr$ |
| Multi-Subspace | ✓ | ✓ | ✗ |
| Adaptive-Subspace-Dimension | ✓ | ✗ | ✗ |
| Adaptive-Subspace-Updates | ✓ | ✗ | ✗ |
| Pre-Training | ✓ | ✓ | ✗ |
| Fine-Tuning | ✓ | ✓ | ✓ |

## 2 RELATED WORK AND BACKGROUND

**Memory efficient optimizers.** Memory-efficient optimization has been a recent focus of research. Multiple studies have aimed to reduce the memory requirements of gradient statistics in adaptive optimization algorithms (Shazeer & Stern, 2018; Anil et al., 2019). One common approach is quantization, which helps decrease the memory footprint of optimizer states (Li et al., 2024). Additionally, recent advancements have suggested reducing the memory used by weight gradients by integrating the backward operation with the optimizer update (Lv et al., 2023a;b). This characteristic has been leveraged to reduce memory usage during training processes (Gooneratne et al., 2020; Huang et al., 2023; Modoranu et al., 2023).

**Low-rank gradient optimization.** The phenomenon of low-rank gradients naturally arises during the training of neural networks, a subject that has been extensively examined both theoretically and practically, e.g., Zhao et al. (2022); Cosson et al. (2023); Yang et al. (2023). This characteristic low-rank structure gradients has been leveraged to reduce memory usage during training processes Gooneratne et al. (2020); Huang et al. (2023); Modoranu et al. (2023), and results in a reduced computational complexity as compared to standard gradient descent methods.

**Adam optimization.** Arguably, among the most popular optimization methods used for training large language models (LLMs) are the *Adam optimizer* (Kingma & Ba, 2017) and its variant, *AdamW* (Loshchilov & Hutter, 2019), which incorporates weight decay for regularization. However, it is well-established that Adam optimization has higher memory complexity compared to other optimization alternatives. To illustrate this, let us briefly review how the Adam optimization algorithm operates. First, we need to establish some notation. Consider a neural network denoted as $\Phi(\cdot; \boldsymbol{\theta})$, which consists of $L$ layers and is parameterized by $\boldsymbol{\theta} \triangleq \left[ \mathbf{W}_1^{d_1 \times d_0}, \dots, \mathbf{W}_{L-1}^{d_{L-1} \times d_{L-2}}, \mathbf{W}_L^{d_L \times d_0^{L-1}} \right]$. Here, $\mathbf{W}_i$ represents the weights tensor parameters associated with the $i$-th layer, for $i \in [L]$. In the following, let $t \in \mathbb{N}$ represent the $t$-th step of the Adam optimization algorithm. Then, we recall that the single update step in Adam is given by,

Specifically, at time step $t$, $\mathbf{G}_t$ denotes the backpropagated gradient matrix, i.e., $\nabla \Phi (\boldsymbol{\theta}_{t-1})$. The exponentially weighted moving averages of the first and second moments are denoted by $\mathbf{M}_t$ and $\mathbf{V}_t$, respectively, with their bias-corrected counterparts given by $\hat{\mathbf{M}}_t$ and $\hat{\mathbf{V}}_t$. The AdamW optimizer updates the model parameters at step $t$ according to the rule, $\boldsymbol{\theta}_t = \boldsymbol{\theta}_{t-1} - \alpha \left( \frac{\hat{\mathbf{M}}_t}{\sqrt{\hat{\mathbf{V}}_t} + \epsilon} + \lambda \boldsymbol{\theta}_{t-1} \right)$, where $\lambda \geq 0$ is the weight decay rate (for Adam $\lambda = 0$), and all operations are performed element-wise. In this equation, $\beta_1$ and $\beta_2$ control the decay rates for the moving averages of the moments, $\alpha$ is the

| Adam (single update step) |
| :--- |
| $\mathbf{G}_t = \nabla \Phi (\boldsymbol{\theta}_{t-1})$, |
| $\mathbf{M}_t = \beta_1 \mathbf{M}_{t-1} + (1 - \beta_1) \mathbf{G}_t$, |
| $\mathbf{V}_t = \beta_2 \mathbf{V}_{t-1} + (1 - \beta_2) \mathbf{G}_t^2$, |
| $\hat{\mathbf{M}}_t = \mathbf{M}_t / (1 - \beta_1^t)$, |
| $\hat{\mathbf{V}}_t = \mathbf{V}_t / (1 - \beta_2^t)$, |
| $\boldsymbol{\theta}_t = \boldsymbol{\theta}_{t-1} - \alpha \hat{\mathbf{M}}_t / \left( \sqrt{\hat{\mathbf{V}}_t} + \epsilon \right)$. |

learning rate, and $\epsilon$ is a small constant to avoid division by zero. Notably, since Adam/W requires storing both $\mathbf{M}_t$ and $\mathbf{V}_t$ at each time step, it incurs an additional memory footprint of $2mn$.

While existing approaches (Zhao et al., 2024b; Vyas et al., 2024; Okewu et al., 2020) focus on low-rank approximations of the first and second moments with the goal of reducing memory requirements, we propose to approximate the gradients by a low-rank factorization. Consequently, in our scheme the moments are integrally constrained onto this reduced dimension, and thus we gain both benefits.

## 3 METHOD AND MAIN RESULTS

### 3.1 THEORETICAL MOTIVATION: GRADUALLY GRADIENT RANK VANISHING

As mentioned in the introduction, a few recent empirical results (e.g., (Jaiswal et al., 2024b; Zhao et al., 2024a; Lialin et al., 2023a)), demonstrate that the gradients, when training or fine-tuning LLM's, are "approximately low-rank". As an example, this phenomenon can be observed in Figure 2, where it is evident that the squared norm of the gradient's singular values decay to zero exponentially fast.

As hinted above, this phenomenon is only true in the approximate sense; roughly speaking, only very few eigenvalues hold almost all the information captured by the gradient. Accordingly, a low-rank matrix approximates the underlying gradient up to a negligible approximation error. The practical implication is that while the weight matrices are not necessarily low-rank, training certain high-rank layers with low-rank based-gradient updates is possible. To make the above discussion precise, consider the following definition for approximate low-rank matrices.

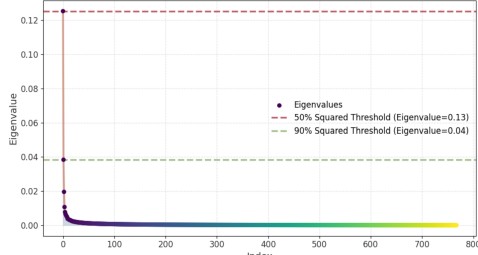

Figure 2: The figure illustrates the exponential decay of eigenvalues in the MLP layer's gradient, at the first iteration of fine-tuning RoBERTa-Base (Liu, 2019) model, on MRPC task, from GLUE (Wang et al., 2019). Notably, the red line indicates that 50% of the gradient information (in terms of squared norm ratio) is captured by the first eigenvalue, while the green line shows that 90% is contained within the first two eigenvalues.

**Definition 1** (Approximate low-rank matrix). *A matrix* $\mathbf{A} \in \mathbb{R}^{n \times m}$ *is called* $(\eta, \varepsilon)$*-approximately rank-$r$, if there exist* $\eta \in [0, 1)$, $\varepsilon > 0$, *and a matrix* $\mathbf{A}_{\mathsf{app},r} \in \mathbb{R}^{n \times m}$ *with* $\mathsf{rank}(\mathbf{A}_{\mathsf{app},r}) = r$ *and* $r < \min\{n, m\}$, *such that,*

$$\| \mathbf{A} - \mathbf{A}_{\mathsf{app},r} \|_F \leq \eta \cdot \| \mathbf{A} \|_F + \varepsilon. \tag{1}$$

As it turns out, it can be shown (see, e.g., (Golub & Van Loan, 2013)) that the optimal $\mathbf{A}_{\mathsf{app},r}$ minimizing the approximation error in the left-hand-side of equation 1, can be obtained by applying an SVD on $\mathbf{A}$ and then retaining only the top $r$ singular values and their corresponding singular vectors. Mathematically, we have $\mathbf{A}_{\mathsf{app},r} = \sum_{i=1}^{r} \sigma_i \mathbf{u}_i \mathbf{v}_i^{\top}$, where $\{\sigma_i\}_i$ are the singular values of $\mathbf{A}$, and $\{\mathbf{u}_i\}_i$ and $\{\mathbf{v}_i\}$ are the corresponding left and right singular vectors, respectively. The approximation error is in turn given by $\|\mathbf{A} - \mathbf{A}_{\mathsf{app},r}\|_F^2 = \sum_{i=r+1}^{\min\{m,n\}} \sigma_i^2$. This construction gives an $(\eta_{\mathbf{A}}, 0)$-approximately rank-$r$ matrix, with the minimal $\eta_{\mathbf{A}}$ possible.

Recently in (Zhao et al., 2024a), the structure of the gradient for a wide family of nonlinear networks known as "reversible networks" (Tian et al., 2021) was studied,[1] defined as follows.

**Definition 2.** *(Reversibility* $(Tian\,et\,al.,\,2021)$*) A layer l is reversible if there is a* $\mathbf{G}_\ell(\boldsymbol{x}; \boldsymbol{\theta}) \in \mathbb{R}^{n_\ell \times n_{\ell-1}}$ *so that the pre-activation at layer $\ell$ satisfies* $\tilde{\boldsymbol{f}}_\ell(\boldsymbol{x}; \boldsymbol{\theta}) = \mathbf{G}_\ell(\boldsymbol{x}; \boldsymbol{\theta})\tilde{\boldsymbol{f}}_{\ell-1}(\boldsymbol{x}; \boldsymbol{\theta})$ *and backpropagated gradient after nonlinearity* $\tilde{\boldsymbol{g}}_{\ell-1} = \mathbf{G}_\ell^{\top}(\boldsymbol{x}; \boldsymbol{\theta})\mathbf{P}_\ell^{\top}(\boldsymbol{x}; \boldsymbol{\theta})\tilde{\boldsymbol{g}}_\ell$*, for some matrix* $\mathbf{P}_\ell(\boldsymbol{x}; \boldsymbol{\theta}) \in \mathbb{R}^{n_\ell \times n_\ell}$*. A network is reversible if all layers are.*

For simplicity of notation, we use $\mathbf{G}_t^\ell$ to denote $[\mathbf{G}_\ell(\boldsymbol{x}; \boldsymbol{\theta})]_t$, where $t \in \mathbb{N}$ is the iteration index in the optimization process. Furthermore, when it is clear from the context, we omit the layer index $\ell$ from our notations. Assuming reversibility and SGD weight update (i.e., $\mathbf{W}_t = \mathbf{W}_{t-1} + \alpha \mathbf{G}_{t-1}$), it is shown in Zhao et al. (2024a) that for both $\ell_2$ and cross entropy losses, the gradient is of the structure form $\mathbf{G} = \frac{1}{N} \sum_{i=1}^{N} (\mathbf{A}_i - \mathbf{B}_i W \mathbf{C}_i)$, where $N$ is the batch size, $\{\mathbf{A}_i\}_{i=1}^N$ are input-dependent matrices, and $\{\mathbf{B}_i, \mathbf{C}_i\}_{i=1}^N$ are certain positive semi-definite (PSD) matrices. Furthermore, it was proven that if the gradient $\mathbf{G}_t$, has the above structure for all $t \geq \mathsf{t}_0$, for some $\mathsf{t}_0 \in \mathbb{N}$, then, the stable rank $\mathrm{sr}(\mathbf{G}_t) \triangleq \frac{\|\mathbf{G}_t\|_F}{\|\mathbf{G}_t\|_2}$ satisfies,

$$\mathrm{sr}(\mathbf{G}_t) \leq \mathrm{sr}\left(\mathbf{G}_{\mathsf{t}_0}^{\|}\right) + \left(\frac{1 - \eta\lambda_2}{1 - \eta\lambda_1}\right)^{2(t-\mathsf{t}_0)} \left\|\mathbf{G}_{\mathsf{t}_0} - \mathbf{G}_{\mathsf{t}_0}^{\|}\right\|_F^2 \Big/ \left\|\mathbf{G}_{\mathsf{t}_0}^{\|}\right\|_2^2,$$

where $\mathbf{S} \triangleq \frac{1}{N} \sum_{i=1}^N \mathbf{C}_i \otimes \mathbf{B}_i$, $\lambda_1 < \lambda_2$ denote its two smallest distinct eigenvalues, and $\mathbf{G}_{\mathsf{t}_0}^{\|}$ is the projection of $\mathbf{G}_{\mathsf{t}_0}$ onto the minimal eigenspace $\mathcal{V}_1$ of $\mathbf{S}$ that corresponds to $\lambda_1$. Accordingly, as $t \to \infty$, we get that the final stable rank is upper bounded by $\mathrm{sr}(\mathbf{G}_{\mathsf{t}_0}^{\|})$. Under the same gradient structure assumption and for the vanilla settings of the SGD weights update Battash et al. (2024), we were able to prove the following stronger result. We prove that the approximated stable rank of the gradients approach one as the training process progresses. To state this result, we need to make a few notations. Let $\mathbf{G}_t = \mathbf{U}_t \Sigma_t \mathbf{V}_t^{\top}$ be the SVD decomposition of $\mathbf{G}_t$, and let $\mathbf{P}_t(l, r) = \mathbf{U}[:, l : r]_t \mathbf{U}[:, l : r]_t^{\top}$ be the corresponding projection matrix. When clear for the context, we omit the index $l$ and use $\mathbf{P}_t(r) \equiv \mathbf{P}_t(l = 1, r)$. We have the following result.

**Lemma 1** (Asymptotically rank-one). *If a neural network is trained using vanilla SGD, then the following holds for the gradient of a reversible layer at iteration $t$,*

$$\kappa(t) \triangleq \frac{\|\mathbf{G}_t - \mathbf{P}_t(1)\mathbf{G}_t\|_F^2}{\|\mathbf{G}_t\|_F^2} \leq O(C^{-t}),$$

*for some constant $C > 1$.*

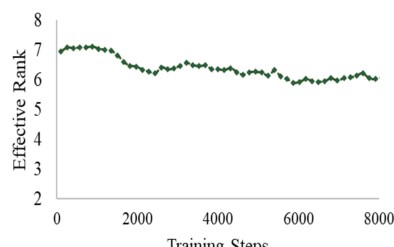

Figure 3: The figure presents the effective rank (see Section 4) measured after every 100 update steps on the RTE dataset, from GLUE (Wang et al., 2019).

The above result implies that $\mathbf{G}_t$ approaches its rank-one approximation $\mathbf{P}_t(1)\mathbf{G}_t$, as the iteration number increases, namely, $\mathbf{G}_t$ becomes rank-one. The proof of Lemma 1 is relegated to Section B.

Finally, in Fig. 3 and Fig. 4, we demonstrate that for a large language model (RoBERTa-base Liu (2019)), which contains also non reversible layers, the rank decay evolves as a function of the number of update steps, in a fine-tuning task.

---

[1]It can be shown that this family includes many different kinds of layers, such as, linear layers (MLP and Conv.), and (leaky) ReLU non-linearity.

## 3.2 ADAPTIVE-LOW-RANK SUBSPACE SELECTION

As aforesaid, our training strategy is self-adapting by updating the low-rank projection subspace for the gradient of each layer, leveraging the phenomena raised by Lemma 1. The most natural but computationally expensive way to search for the subspace on which a gradient has the highest variance is by applying the truncated SVD (Eckart & Young, 1936), with a truncated rank that preserves the predefined fraction of information. In this section, we first utilize an efficient method to approximate the gradient, which is more practical than the expansive truncated SVD approximation. We then use this method to propose an algorithm that identifies the most suitable projection subspace with the smallest possible span/rank that maintains the given information threshold.

### 3.2.1 IDENTIFYING PROJECTING SUBSPACE BY POWER ITERATION

Consider a matrix $\mathbf{A} \in \mathbb{R}^{n \times m}$. Finding its "best" low-rank approximation can be framed as the following optimization problem $\min_{\mathbf{Q},\mathbf{U}} \left\| \mathbf{A} - \mathbf{Q}\mathbf{U}^\top \right\|_F^2$, where $\mathbf{Q} \in \mathbb{R}^{n \times r}$ and $\mathbf{U} \in \mathbb{R}^{m \times r}$. As discussed above, the matrix $\mathbf{A}_{\mathsf{app},r} = \mathbf{Q}\mathbf{U}^\top$ represents the $(\eta, \varepsilon)$ rank-$r$ approximation of $\mathbf{A}$. Computing full SVD for large matrices is computationally intensive and memory-demanding. To address these challenges, we detail Algorithm 1 (Halko et al., 2010), an efficient technique for producing a "good" proxy for the optimal low-rank approximation of $\mathbf{A}$.

Algorithm 1 is designed to solve the optimization problem $\arg\min_{\mathbf{Q} \in \mathbb{R}^{n \times r}} \left\| \mathbf{A} - \mathbf{Q}\mathbf{Q}^\top \mathbf{A} \right\|_F$, and approximates the matrix $\mathbf{A}$ as $\mathbf{A}_{\mathsf{app},r} \approx \mathbf{Q}\mathbf{Q}^\top \mathbf{A}$. The computational complexity of the SSRF algorithm is dominated by two main operations: the matrix multiplication $\mathbf{A}\Omega$, whose computational complexity is $O(mnr)$, and the QR decomposition of the resulting matrix $\mathbf{Y}$, requiring $O(mr^2)$. Therefore, the total complexity is $O(mnr + mr^2)$.

---
**Algorithm 1** Subspace selection via randomized range finder (SSRF)

---
**Inputs:** Matrix $\mathbf{A} \in \mathbb{R}^{n \times m}$, target rank $r \leq \min\{n, m\}$.
**Initialization:** $\Omega \in \mathbb{R}^{m \times r} \sim \mathbf{N}(0, 1)$
$\mathbf{Y} \leftarrow \mathbf{A}\Omega$
Construct $\mathbf{Q} \in \mathbb{R}^{m \times r}$ using the QR decomposition of $\mathbf{Y}$
**Return:** $\mathbf{Q}$

---

In scenarios where $r \ll m, n$, this complexity simplifies to $O(mnr)$, which is significantly more efficient than using SVD, whose complexity is $O(\min(mn^2, m^2n))$. This low complexity makes the SSRF preferable for extracting leading singular vectors in large-scale data settings.

### 3.2.2 FAST ADAPTIVE LOW-RANK APPROXIMATION

It is clear from Definition 1 that a larger value of the rank $r$ increases the computational operations and induces a higher memory footprint, while a smaller value of $r$ will result in a non-negligible approximation error. To mitigate this, we suggest Algorithm 2, an adaptive procedure for rank selection that adjusts the value of $r$ to preserve at least a certain fraction of the information in $\mathbf{A}$. Specifically, at start we set $r_0 = r_{\min}$, and then find the best $\mathbf{A}_{\mathsf{app},r} \approx \mathbf{Q}\mathbf{Q}^\top \mathbf{A}$ using SSRF. At each time step $t$, given the current rank $r_t$, we compute the approximation error as,

$$\eta_t = \frac{\|\mathbf{A} - \mathbf{A}_{\mathsf{app},r_t}\|_F^2}{\|\mathbf{A}\|_F^2}$$
$$= \frac{\sum_{i=r_t+1}^{\min\{n,m\}} \sigma_i(\mathbf{A})^2}{\sum_{i=1}^{n} \sigma_i(\mathbf{A})^2}. \qquad (2)$$

---
**Algorithm 2** Information-based adaptive subspace selection (IASS)

---
**Inputs:** Matrix $\mathbf{A} \in \mathbb{R}^{n \times m}$, minimum rank $r_{\min} \geq 1$, maximum rank $r_{\max} \ll \min\{n, m\}$, and information threshold $\eta_{\mathsf{th}} \in (0, 1)$.
**Initializations:** $t \leftarrow 0, r_0 \leftarrow r_{\min}, \eta_0 \leftarrow \eta_{\mathsf{th}} + 1$.
**while** $r \geq 1$ and $r_{\max} - r_t \geq 1$ **do**
    $\mathbf{Q} \leftarrow \mathrm{SSRF}(\mathbf{A}, r_t)$          {Call Algorithm 1}
    $\mathbf{A}_{\mathsf{app},r_t} \leftarrow \mathbf{Q}\mathbf{Q}^\top \mathbf{A}$.
    $\eta_t \leftarrow \frac{\|\mathbf{A} - \mathbf{A}_{\mathsf{app},r_t}\|_F^2}{\|\mathbf{A}\|_F^2}$          $\left\{ \approx \frac{\sum_{i=r_t+1}^{\min\{n,m\}} \sigma_i(\mathbf{G}_t)^2}{\sum_{i=1}^{n} \sigma_i(\mathbf{G}_t)^2} \right\}$
    **if** $\eta_t > \eta_{\mathsf{th}}$ **then**
        $r_{\max} \leftarrow \left\lceil \frac{r_{\max} - r_t}{2} \right\rceil$
    **else if** $\eta_t \leq \eta_{\mathsf{th}}$ **then**
        $r_t \leftarrow \left\lfloor \frac{r_{\max} - r_t}{2} \right\rfloor$
    **end if**
**end while**
**Return:** $\mathbf{Q}, r_t$.

---

We then apply binary search until we find the maximum $r$ for which the corresponding condition still holds: $\eta < \eta_{\mathsf{th}}$. It is important to note that, from the outset, we know a priori that the rank of the approximated gradients is inherently low. Consequently, the initial value of $r_{\max} \ll \min\{m, n\}$ is

small, and the number of iterations required for the search is correspondingly minimal, specifically $O(\log(r_{\max} - r_{\min}))$, obviously $(r_{\max} - r_{\min}) \ll \min\{m, n\}$.

## 3.3 ADAPTIVE LOW-RANK AND MOMENTS GRADIENT OPTIMIZATION

We can now present our main algorithm for adaptive low-rank and moments gradient optimization in Algorithm 3. The mathematical formulation of the weights update rule proposed in this paper is detailed in Appendix A. comprises four main blocks, all contained within an outer loop that terminates once we reach convergence. The role of each block is as follows.

- **Block 1**: We select the (approximated) subspace along the directions of the $r$ largest eigenvectors, using Algorithm 2. The number of orthogonal directions $r$ is determined by the information threshold required to preserve the gradient information, according to equation 2.

- **Block 2**: We transform the first and second gradient moments evaluated during the Adam update steps between the previous and the updated subspace. The main reason for this transformation is because, as will be seen below, in the fourth block, the first and second moments of the gradients are aligned with the previous projected subspace, and thus, a transformation is needed to convert them from the previous subspace to the current one.

- **Block 3**: We tune the parameters during the low-rank update step. The stopping condition triggering the update of the orthogonal projection matrix is based on the convergence of the projected gradient onto the subspace.

- **Block 4**: The actual pre-trained model parameters are updated, using the low-dimensional and memory-efficient projected gradients on the selected subspace.

**Theorem 2** (Convergence of Algorithm 3). *For a loss function $\mathcal{L}$, and given architecture $\Phi$, suppose that the compositions of $f \equiv \mathcal{L}(\Phi(\cdot))$ is $\beta$-smooth non-convex function that are bounded by some $M \in \mathbb{R}_+$. Let $\mathbf{G}_t^j$ denote the gradient matrix w.r.t. the $j$th reversible layer at time $t \in \mathbb{N}$. Assume that $\|\mathbf{G}_t^j\|_F \leq D$, for all $j \in [L]$ and $t \in \mathbb{N}$, where $D \in \mathbb{R}_+$. Then, for any $\varepsilon > 0$, there exists $\mathsf{C} \in \mathbb{R}_+$ such that for all $\mathsf{T}_N > \frac{\mathsf{C}}{\varepsilon^2}$, it holds that $\frac{1}{\mathsf{T}_N L} \sum_{j=1}^{L} \sum_{i=0}^{N-1} \sum_{t=\mathsf{T}_i}^{\mathsf{T}_{i+1}-1} \left\|\mathbf{G}_t^j\right\|_F^2 \leq \varepsilon$. In particular, Algorithm 3, with vanilla SGD weight update[2], $\varsigma_2 \triangleq \varsigma_{2,i} = \sqrt{1 - \eta_{th}} \cdot \left\|\mathbf{G}_{\mathsf{T}_{i-1}}\right\|_F^2$, where $i$ is the Block 3 entry counter, and learning rate $\alpha < 2/\lambda_{\max}$, achieves an $\varepsilon$-critical point,[3] i.e., $\left\|\mathbf{G}_t^j\right\|_F^2 \leq \varepsilon$, for some $t \in \mathbb{N}$, and any $j \in [L]$.*

Note that, for clarity, we can assume, without loss of generality, that $m \leq n$. In the opposite case, the projection matrix would multiply the gradient from the right side.

The proof of Theorem 2 can be found in Section B. A few important comments are in order. First, to reduce memory usage, we apply in Algorithm 3 a per-layer weight update during backpropagation, as proposed by recent works, see, e.g., Lv et al. (2024). This is in contrast to common optimizers which usually update all weights after backpropagation by storing the full gradients in memory, which could be highly inefficient. Second, note that the Adam update step block in Algorithm 3 can be replaced by any quantized Adam variant, e.g., Li et al. (2017); Chen et al. (2021); Seok & Kim (2021), and as so allows to obtain tasked fine-tuned quantized model or quantized adaptor; this is discussed in more detail below. Finally, we would like to mention here that one can easily apply 4-bit projected gradient updates, as introduced in Q-GaLore (Zhang et al., 2024).

If, after fine-tuning, one wishes to create an adapter (i.e., a parallel low-dimensional LoRA type model) alongside the original model, this can be done efficiently as follows. First, we calculate the training weights gap $\Delta \triangleq \mathbf{W}_{\text{Fine-Tuned}} - \mathbf{W}_{\text{Pretrained}}$, where $\mathbf{W}_{\text{Fine-Tuned}}$ is the model weight at the end of the process, and $\mathbf{W}_{\text{Pretrained}}$ is the original model weight. Then, we find the $r_{\text{Adaptor}} \triangleq \text{rank}(\Delta)$, using some matrix ranking algorithm, and finally, we solve $\min_{\mathbf{A} \in \mathbb{R}^{n \times r_{\text{Adaptor}}}, \mathbf{B} \in \mathbb{R}^{r_{\text{Adaptor}} \times m}} \|\Delta - \mathbf{AB}\|_F^2$, using any optimization algorithm (e.g., gradient descent). Note that any solution to this optimization problem is a global optimum (Kawaguchi, 2016).

---

[2] We focus on SGD for the simplicity (as is standard practice in related literature, e.g., (Zhao et al., 2024a)).

[3] Also known as $\varepsilon$-stationary, see, e.g., (Cosson et al., 2023).

---

**Algorithm 3** Adaptive low-rank and moments gradient (AdaRankGrad)

---

**Input:** A Layer $\mathbf{W} \in \mathbb{R}^{n \times m}$ from $\boldsymbol{\theta}$, dataset $\mathcal{D}$, loss function $\mathcal{L}$, information thershold $0 < \eta_{\text{th}} < 1$, initial rank $r_{\text{init}}$, maximal rank $r_{\max}$, learning rate $\alpha$, and small numbers $\varsigma_1, \varsigma_2 > 0$.
**Initialization:** $t = 0$.
Sample batch $B \longleftarrow \{x_i, y_i\}_{i=1}^{|B|} \sim \mathcal{D}$
Compute batch gradient $\mathbf{G}_t \leftarrow \sum_{i=1}^{|B|} \frac{\partial}{\partial \mathbf{W}} \mathcal{L}(\Phi(x_i, \boldsymbol{\theta}), y_i)$         {For unsupervised $\mathcal{L}(\Phi(x_i, \boldsymbol{\theta}))$}
**while** $\|\mathbf{G}_t\|_F > \varsigma_1$ **do**

---

   **Block 1: Adaptive subspace selection**
   $\mathbf{Q}_t, r_t \leftarrow \text{IASS}(\mathbf{G}_t, r_{\text{init}}, r_{\max}, \eta_{\text{th}})$    {Approximated $r_t$-dimension subspace: $Q_t^{r_t \times n}$ projected matrix}

---

   **Block 2: Moments subspaces transformation**
   $\mathbf{R}_t^{r_t \times r_{t-1}} \leftarrow \mathbf{Q}_t^\top \mathbf{Q}_{t-1}$ if $t \geq 1$, else $\mathbf{0}^{r_t \times r_{t-1}}$
   $\mathbf{M}_t^{r_t \times m} \leftarrow \mathbf{R}_t \mathbf{M}_{t-1}$, if $t \geq 1$, else $\mathbf{0}^{r_t \times m}$         {$1^{st}$-order moment}
   $\mathbf{V}_t^{r_t \times m} \leftarrow \mathbf{R}_t \mathbf{V}_{t-1}$, if $t \geq 1$, else $\mathbf{0}^{r_t \times m}$         {$2^{st}$-order moment}

---

   **Block 3: Low-rank optimization**
   $\hat{\mathbf{G}}_t \leftarrow \mathbf{Q}_t^\top \mathbf{G}_t$         {Projected gradient on the approximated $r_t$-dimension subspace}
   **while** $\|\hat{\mathbf{G}}_t\|_F > \varsigma_2$ **do**

---

      **Block 4: Adam update step**
      $\mathbf{M}_t \longleftarrow \beta_1 \mathbf{M}_t + (1 - \beta_1) \hat{\mathbf{G}}_t$
      $\mathbf{V}_t \longleftarrow \beta_2 \mathbf{V}_t + (1 - \beta_2) \hat{\mathbf{G}}_t^2$
      $\hat{\mathbf{M}}_t \longleftarrow \mathbf{M}_t / (1 - \beta_1^t)$
      $\hat{\mathbf{V}}_t \longleftarrow \mathbf{V}_t / (1 - \beta_2^t)$
      $\mathbf{W}_t \longleftarrow \mathbf{W}_t - \alpha \mathbf{Q}_t \hat{\mathbf{M}}_t / \left( \sqrt{\hat{\mathbf{V}}_t} + \epsilon \right)$

---

      $t \leftarrow t + 1$
      Sample batch $B \longleftarrow \{x_i, y_i\}_{i=1}^{|B|} \sim \mathcal{D}$
      Compute batch gradient $\mathbf{G}_t \leftarrow \sum_{i=1}^{|B|} \frac{\partial}{\partial \mathbf{W}} \mathcal{L}(\Phi(x_i, \boldsymbol{\theta}), y_i)$
      $\hat{\mathbf{G}}_t \leftarrow \mathbf{Q}_t^\top \mathbf{G}_t$
   **end while**

---

   **end while**         {Exit by convergence criteria could alternatively be defined by the number of epochs}
   **Return** $\mathbf{W}_t$

---

## 4 EXPERIMENTS

In this section, we test the performance of our algorithm on real-world datasets. Before discussing the setup we rely on in our experiments, we define four measures for memory usage reduction in the rank-adaptive projection matrices. Specifically, for the $j$th layer, we define *the effective layer-gradient-rank*, by, $\mathcal{R}_{\text{adap}}^j \triangleq \frac{\sum_{t=0}^{\mathsf{T}-1} \mathsf{R}_t^j}{\mathsf{T}}$, where $\mathsf{R}_t^j$ is the rank of the $j$th layer projection matrix at time $0 \leq t \leq \mathsf{T} - 1$, for some $\mathsf{T} \in \mathbb{N}$, and accordingly the *total weighted-average effective rank* is defined as $\mathcal{R}_{\text{adap}} \triangleq \frac{\sum_{j=1}^L \sum_{t=0}^{\mathsf{T}-1} \mathsf{R}_t^j (d_j + d_{j+1})}{\mathsf{T} \cdot \sum_{j=0}^L (d_j \cdot d_{j+1})}$. Following that, we define the average per-layer reduction in memory footprint, when compared to the non-adaptive low-rank fine-tuning, by $\mathcal{M}_{\text{red}}^j \triangleq (\bar{\mathsf{R}}^j - \mathcal{R}_{\text{adap}}^j) \cdot (d_j + d_{j+1})$, where $\bar{\mathsf{R}}^j$ is the time-independent non-adaptive rank (such as in Galore). Finally, we define the total memory reduction by $\mathcal{M}_{\text{red}} \triangleq \sum_{j=1}^L (\bar{\mathsf{R}}^j - \mathcal{R}_{\text{adap}}^j) \cdot (d_j + d_{j+1})$.

**Fine-tuning on GLUE benchmark.** We evaluate our model on the GLUE benchmark (Wang et al., 2019) by fine-tuning the pre-trained Roberta-base model Liu (2019) on 8 tasks. We compare the results against full fine-tuning, LoRA, and GaLore methods and present them in Table 2. We report the overall (matched and mismatched) accuracy for MNLI, Matthew's correlation for CoLA, Pearson correlation for STS-B, F1-score for MRPC, and accuracy for other tasks. As can be seen, our method improves the accuracy of fine-tuning while consuming less training memory on average. Fig. 4 shows the average memory reduction measured at the end of each epoch (blue) and the corresponding effective rank (green). Empirical analysis of the hyperparameters appears in Appendix D.

Table 2: Evaluating AdaRankGrad comparing to state-of-the-art memory-efficient fine-tuning methods on GLUE benchmark using pre-trained RoBERTa-Base. For AdaRankGrad, we present accuracy results with average effective rank.

| Model | Memory | CoLA | STS-B | MRPC | RTE | SST2 | MNLI | QNLI | QQP |
|---|---|---|---|---|---|---|---|---|---|
| Full Fine-Tuning | 747M | 62.24 | 90.92 | 91.30 | 79.42 | 94.57 | 87.18 | 92.33 | 92.28 |
| LoRA (rank=4) | 257M | 61.38 | 90.57 | 91.07 | 78.70 | 92.89 | 86.82 | 92.18 | **91.29** |
| GaLore (rank=4) | 253M | 60.35 | 90.73 | 92.25 | 79.42 | 94.0 | 87.0 | 92.24 | 91.06 |
| AdaRankGrad (Initial rank=4) | 202M | **61.4**(3.71) | **90.97**(3.79) | **92.6**(3.72) | **81.23**(3.65) | **94.8**(3.91) | 86.6(3.69) | **92.5**(3.9) | 90.4(3.67) |
| LoRA (rank=8) | 264M | 61.83 | 90.80 | 91.90 | 79.06 | 93.46 | 86.94 | 92.25 | **91.22** |
| GaLore (rank=8) | 257M | 60.06 | 90.82 | 92.0 | 79.78 | 94.38 | 87.17 | 92.2 | 91.11 |
| AdaRankGrad (Initial rank=8) | **237M** | **62.0**(6.41) | **90.89**(7.77) | **93.2**(5.33) | **81.23**(6.64) | **94.80**(5.69) | 86.5(6.00) | **92.6**(6.54) | 89.7(6.00) |

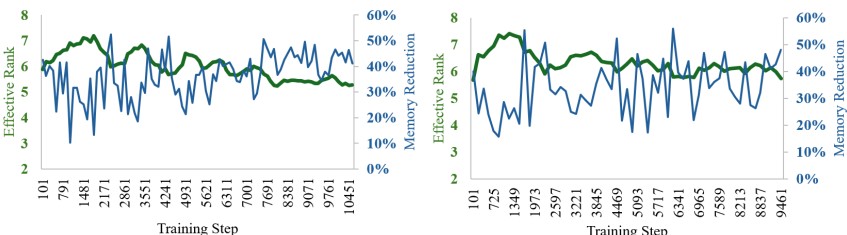

Figure 4: We present the effective rank measured for non-attention layers and corresponding memory reduction for AdaRankGrad trained on MRPC (left panel) and RTE (right panel) datasets from the GLUE benchmark.

**Fine-tuning Geneformer on Biological Omics Tabular Data**   High-throughput omics technologies, such as next-generation sequencing Reis-Filho (2009), allow for the simultaneous measurement of thousands to millions of biological molecules, capturing biological processes at the tissue or single-cell level. Recent studies have aimed to construct foundation models for omics data Cui et al. (2024); Theodoris et al. (2023). Unfortunately, the complexity of omics data makes the use of low-rank optimization methods such as LoRa sub-optimal for foundation omics models, and therefore, training is often based on relatively high ranks (Chen et al., 2024). In this experiment, we demonstrate that the proposed approach can alleviate this problem due to the adaptive low-rank projection of gradients. We conducted a fine-tuning experiment for cell classification, specifically focusing on the Classification of Cardiomyopathy Disease States. We used the *Geneformer gf-12L-30M-i2048* [4] as our pre-trained base model, which is based on the BERT model and pre-trained on 30 million single-cell transcriptomes (Theodoris et al., 2023; Devlin, 2018). Following the setup proposed in (Chen et al., 2024), we fine-tuned our method with a maximal rank of 16 on approximately 93,600 samples for three epochs, with a batch size of 16. We evaluated the model on a test set of size 17,228. Figure 5 presents the Macro-F1 score and Accuracy measured for each epoch. As shown, our method converges faster with a stable improvement over LoRA.

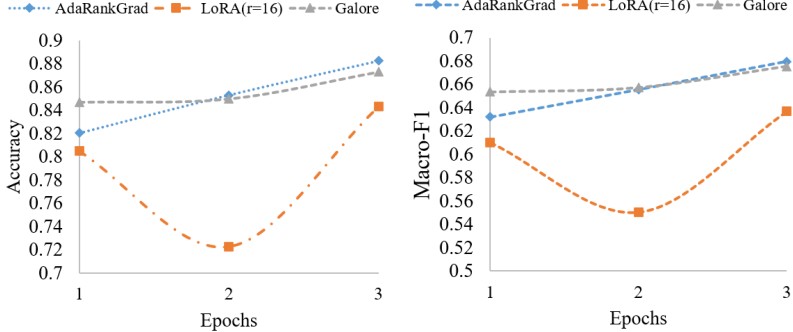

Figure 5: We evaluate AdaRankGrad in the Geneformer fine-tuning task and present Accuracy (left panel) and Macro-1 (right panel) measurements. The adaptive low-rank projections improve the model converges compared to LoRA and Galore methods.

---

[4]A transformer-based model trained on  30 million single-cell transcriptomes for gene classification and in silico perturbation analysis. It has 12 layers with an input size of 2048 genes per cell, designed to understand gene network dynamics. Huggingface: `https://huggingface.co/ctheodoris/Geneformer`

**Pre-training LLAMA on C4 Dataset** Here, we repeated the comparison presented in Zhao et al. (2024a) [Table 2] to evaluate AdaRankGrad performance to the state-of-the-art method, in terms of perplexity and memory usage. We evaluate AdaRankGrad by training large LLaMA-based models on the C4 dataset, a cleaned and massive version of Common Crawl's web corpus (Raffel et al., 2020). This dataset is primarily used for pre-training language models and learning word representations. To closely mimic practical pre-training scenarios, we train on a large dataset without repeating data, scaling model sizes up to 350 million. The results are presented in Table 4.

Table 3: The LLaMA 7B model was pre-trained on the C4 dataset for 120K steps. Validation perplexity and memory usage estimates reported.

| Steps/Tokens | 8-bit GaLore | 8-bit Adam | 8-bit AdaRankGrad |
|---|---|---|---|
| 40K / 5.2B | 17.94 | 18.09 | **17.86** |
| 80K /10.5B | 15.39 | 15.47 | **15.27** |
| 120K /15.7B | 14.95 | **14.83** | 14.87 |
| **Mem** | 18G | 26G | **16.4G** |

Table 4: A comparison of low-rank state-of-the-art algorithms for pre-training LLaMA models of varying sizes on the C4 dataset. We use initial rank $r_{\text{init}} = r$ for AdaRankGrad, w.r.t the $r$ presented in the last table raw. The information threshold $\eta_{\text{th}} = 0.48$ for the three model training. The validation perplexity is presented, along with an estimate of the memory required for the total number of parameters and optimizer states in BF16 format. We used NVIDIA A100 for the three experiments.

| | 60M | 130M | 350M | 1B |
|---|---|---|---|---|
| Full-Rank | 34.06(0.36G) | 25.08(0.76G) | 18.80(2.06G) | 15.56(7.80G) |
| GaLore | 34.88(0.24G) | 25.36(0.52G) | 18.95(1.22G) | 15.64(4.38G) |
| Low-Rank | 78.18(0.26G) | 45.51(0.54G) | 37.41(1.08G) | 142.53(3.57G) |
| LoRA | 34.99(0.36G) | 33.92(0.80G) | 25.58(1.76G) | 19.21(6.17G) |
| ReLoRA | 37.04(0.36G) | 29.37(0.80G) | 29.08(1.76G) | 18.33(6.17G) |
| **AdaRankGrad** | **34.24**(0.206G) | **25.22**(0.497G) | **18.91**(1.106G) | **14.71**(3.62G) |
| Training Tokens | 1.1 B | 2.2 B | 6.4 B | 13.1B |
| $r/d_{\text{model}}$ | 128/256 | 256/768 | 256/1024 | 512/2048 |

## 5 DISCUSSION

In this paper, we present AdaRankGrad, a full-parameters efficient optimization scheme that applies adaptive low-rank updates without relying on a parallel low-rank adapter (LoRA), thus preserving the natural training dynamics. Unlike methods such as LoRA, which rely on parallel adapters, AdaRankGrad enables full parameter fine-tuning while maintaining low memory costs through efficient low-rank optimization updates. Moreover, unlike GaLore, AdaRankGrad leverages the natural phenomenon where the dimensionality of the approximated gradient decreases as training progresses. This allows adaptive updates of the projection subspace only when the gradients have converged to a lower-dimensional space, ensuring that optimization resources are fully utilized—no unnecessary updates occur before or after convergence. As demonstrated in Section 4, this approach results in superior prediction performance. AdaRankGrad offers a unique trade-off between memory efficiency and model performance. While it may slightly increase training time due to the need to determine an optimal rank for projecting subspace updates, these updates are infrequent. The search for the best subspace rank is conducted within a very narrow range, reducing the overall computational cost. To further mitigate this, we proposed an efficient subspace-update technique that significantly reduces the SVD-based subspace calculations by an order of magnitude. In practice, this optimization compensates for the additional time spent in subspace search, making AdaRankGrad a competitive choice compared to methods like GaLore. For future research, we suggest exploring other update steps besides Adam, such as AdaFactor, and studying different efficient algorithms for computing the subspace over which gradients are projected. Another promising avenue would be to investigate algorithms that search for the optimal subspace rank, which preserves a similar fraction of information as the full gradient. Finally, evaluating this method's effectiveness in knowledge editing (Rozner et al., 2024) is an interesting topic for future work.

## 6 ACKNOWLEDGMENTS AND DISCLOSURE OF FUNDING

The work of OL is supported by the MOST grant number 0007341.

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

## A  UPDATE STEP RULE FORMULATION

To describe AdaRankGrad update step rule, we need to establish a few important notations first. For $t \in \mathbb{N}$, let $\rho_t : \mathbb{R}^{m \times n} \to \mathbb{R}^{m \times n}$ be an entry-wise gradient update rule (e.g., Adam, AdamW, AdaFactor, etc.). For $t \in \mathbb{N}$ and layer $j \in [L]$, recall the SVD of the gradient $\mathbf{G}_t^j = \mathbf{U}_t^j \Sigma_t^j (\mathbf{V}_t^j)^\top$, and further define the projection matrices $\mathbf{P}_t^j (r_t^j) = \text{SSRF}(\mathbf{U}_t^j[:,:r_t^j]^\top), \mathbf{R}_t^j (r_t^j) = \text{SSRF}\left(\mathbf{V}_t^j[:,:r_t^j]\right)$, where for a given threshold $0 < \eta_{\text{th}} \leq 1$, we let $r_t^j \triangleq \sup\{r \in \mathbb{N} : \eta_{\text{th}} \cdot \|\mathbf{G}_t^j\|_F^2 - \|\mathbf{G}_t^j - \mathbf{P}_t^j(r)\mathbf{G}_t^j\|_F^2 \geq 0\}$, computed using Algorithm 2. Next, for $j \in [L]$, and some $\varsigma > 0$, we let $\{\mathsf{T}_\ell^j\}_{\ell \geq 0}$, with $\mathsf{T}_0^j = 0$, denote the monotone sequence of integers, for which $\|(\mathbf{P}_{\mathsf{T}_i}^j)^\top (r_{\mathsf{T}_i^j}^j)\mathbf{G}_{\mathsf{T}_i^j}^j \mathbf{R}_{\mathsf{T}_i^j}^j (r_{\mathsf{T}_i^j}^j)\|_F \leq \varsigma$; under certain conditions that we list below, it is shown in Cosson et al. (2023)[Thm. 3.4] (as well as in Zhao et al. (2024a)[Theorem 3.8], for reversible layers) that $\mathsf{T}_i$'s are finite. Then, for $i \in \mathbb{N}$, and $t \in [\mathsf{T}_i + 1, \mathsf{T}_{i+1}]$, the AdaRankGrad weight update is given by,

$$\mathbf{W}_t^j = \mathbf{W}_{\mathsf{T}_i}^j + \sum_{\ell = \mathsf{T}_i + 1}^{t} \eta_t \mathbf{P}_{\mathsf{T}_i}^j (r_{\mathsf{T}_i}^j) \rho_\ell \left((\mathbf{P}_{\mathsf{T}_i}^j (r_{\mathsf{T}_i}))^\top \mathbf{G}_\ell^j \mathbf{R}_{\mathsf{T}_i^j}^j (r_{\mathsf{T}_i^j}^j)\right) (\mathbf{R}_{\mathsf{T}_i}^j (r_{\mathsf{T}_i^j}^j))^\top,$$

for $j \in [L]$, where $\eta_t$ is the learning rate at time $t$ adapted by $\rho_t$, and $\mathbf{W}_{\mathsf{T}_0} = \mathbf{W}_0$ is a pre-trained model, in a case of fine-tuning task, or a given weights initialization, in case of pre-training task. In the sequel, we let $\eta \triangleq \eta_0$ denote the initial learning rate, and for simplicity of notation, when clear from the context, we drop the dependency of the various notations on the layer index $j$.

## B  PROOFS

### B.1  PROOF OF LEMMA 1

In this section, we prove Lemma 1. Consider the SVD decomposition of the gradient $\mathbf{G}_t = \mathbf{U}_t \Sigma_t \mathbf{V}_t^\top$, at iteration $t$. For any natural number $r < n$, we denote $\mathbf{H}_t^{n \times r}(r) = \mathbf{U}[:, 1:r]$. For simplicity of notation, let us denote $\mathbf{P}_t(r) = \mathbf{H}_t(r)\mathbf{H}_t^\top(r)$, where $\mathbf{P}_t(r)$ is an orthogonal projection matrix, i.e., $\mathbf{P}_t^\top(r)\mathbf{P}_t(r) = \mathbf{P}_t(r)$, and $\mathbf{P}_t(r) = \mathbf{P}_t^\top(r)$. Without the loss of generality, we assume that at $t = 0$, the rank of $\mathbf{G}_0$ is such that $\text{rank}(\mathbf{G}_0) > r$. For reversible networks, it was shown in (Zhao et al., 2024a)[Theorem 3.2] that the gradients have the form $\mathbf{G}_t = \frac{1}{N} \sum_{i=1}^N (\mathbf{A}_i - \mathbf{B}_i \mathbf{W}_t \mathbf{C}_i)$, with constant matrices $\{\mathbf{A}_i\}_i$, and PSD matrices $\{\mathbf{B}_i, \mathbf{C}_i\}_i$, for $t \geq \mathsf{t}_0$, where $\mathsf{t}_0 \in \mathbb{N}$. Recall that in the vanilla SGD weight update, we have $\mathbf{W}_t = \mathbf{W}_{t-1} + \eta \mathbf{G}_{t-1}$. Let $\mathbf{S} \triangleq \frac{1}{N} \sum_{i=1}^N \mathbf{C}_i \otimes \mathbf{B}_i$, and $\lambda_1 < \lambda_2$ be its two smallest distinct eigenvalues. In order to prove our result we rely on several results and arguments in the proof of Lemma 3.3 in (Zhao et al., 2024a). Specifically, let $\mathbf{G}_{\mathsf{t}_0}^{\|}$ be the projection of $\mathbf{G}_{\mathsf{t}_0}$ onto the minimal eigenspace $\mathcal{V}_1$ of $S$ corresponding to $\lambda_1$. By assumption, we know that the rank of $\mathbf{G}_{\mathsf{t}_0}^{\|}$ is $L$, and its SVD is $\mathbf{G}_{\mathsf{t}_0}^{\|} = \sum_{l=1}^L c_l z_l y_l^\top$, where $\{z_l\}_{l=1}^L$ and $\{y_l\}_{l=1}^L$ are the orthonormal unit vectors, and $\{c_l\}_{l=1}^L$ are the corresponding singular values. Thus, it was in (Zhao et al., 2024a) that,

$$g_0^{\|} = \text{vec}\left(\mathbf{G}_{\mathsf{t}_0}^{\|}\right) = \sum_{l=1}^L c_l (y_l \otimes z_l) \triangleq \sum_{l=1}^L c_l v_l,$$

with unit vector $v_l \triangleq y_l \otimes z_l \in \mathcal{V}_1$. It is clear that

$$v_l^\top v_{l'} = \left(y_l^\top \otimes z_l^\top\right)\left(y_{l'} \otimes z_{l'}\right)$$
$$= \left(y_l^\top y_{l'}\right)\left(z_l^\top z_{l'}\right)$$
$$= \mathbb{I}(l = l'),$$

where $\mathbb{I}$ is the indicator function. Using this, it was finally shown in (Zhao et al., 2024a) that,

$$\|\mathbf{G}_t\|_2 = \max_{\|y'\|_2 = 1, \|z'\|_2 = 1} z'^\top \mathbf{G}_t y'$$
$$\geq \max_l z_l^\top \mathbf{G}_t y_l = \max_l (y_l \otimes z_l)^\top g_t = \max_l v_l^\top (1 - \eta S)^t g_0 = (1 - \eta \lambda_1)^t \max_l v_l^\top g_0.$$

Using the above results from (Zhao et al., 2024a), we can now see that,

$$
\begin{aligned}
\|\mathbf{G}_t\|_2 &= \max_{\|\boldsymbol{y}\|_2=1,\|\boldsymbol{z}\|_2=1} \boldsymbol{z}'^{\top} \mathbf{G}_t \boldsymbol{y} \\
&= \max_{\|\boldsymbol{y}\|_2=1,\|\boldsymbol{z}\|_2=1} (\boldsymbol{y} \otimes \boldsymbol{z})^{\top} g_t \\
&= \max_{\|\boldsymbol{y}\|_2=1,\|\boldsymbol{z}\|_2=1} (\boldsymbol{y} \otimes \boldsymbol{z})^{\top} (1 - \eta S)^t g_0 \\
&= \left(\mathbf{U}_0[:,1]^{\top} \otimes \mathbf{V}_0[:,1]\right)^{\top} (1 - \eta S)^t g_0 \\
&= \tilde{\boldsymbol{v}}^{\top} (1 - \eta S)^t g_0 \\
&= (1 - \eta \lambda_1)^t \tilde{\boldsymbol{v}}^{\top} g_0.
\end{aligned}
$$

Thus, we have

$$
(1 - \eta \lambda_1)^t \tilde{\boldsymbol{v}}^{\top} g_0 \geq (1 - \eta \lambda_1)^t \max_l \boldsymbol{v}_l^{\top} g_0,
$$

or equivalently

$$
(1 - \eta \lambda_1)^t \tilde{\boldsymbol{v}}^{\top} g_0 \geq (1 - \eta \lambda_1)^t \left\|g_0^{\|}\right\|_2. \tag{3}
$$

Now,

$$
\begin{aligned}
\kappa(t) &\triangleq \frac{\|\mathbf{G}_t - \mathbf{P}_t(1)\mathbf{G}_t\|_F^2}{\|\mathbf{G}_t\|_F^2} \\
&= \frac{\|\mathbf{G}_t\|_F^2 - \|\mathbf{G}_t\|_2^2}{\|\mathbf{G}_t\|_F^2} \\
&\leq \frac{\|\mathbf{G}_t\|_F^2 - \|\mathbf{G}_t\|_2^2}{\|\mathbf{G}_t\|_2^2} = \frac{\|\mathbf{G}_t\|_F^2}{\|\mathbf{G}_t\|_2^2} - 1 \\
&\leq \frac{(1 - \eta \lambda_1)^{2t} \left\|g_0^{\|}\right\|_2^2}{\|\mathbf{G}_t\|_2^2} + \frac{(1 - \eta \lambda_2)^{2t} \left\|g_0^{\perp}\right\|_2^2}{\|\mathbf{G}_t\|_2^2} - 1 \\
&\leq \underbrace{\frac{(1 - \eta \lambda_1)^{2t} \left\|g_0^{\|}\right\|_2^2}{(1 - \eta \lambda_1)^{2t} \tilde{\boldsymbol{v}}^{\top} g_0}}_{\leq 1} + \frac{(1 - \eta \lambda_2)^{2t} \left\|g_0^{\perp}\right\|_2^2}{(1 - \eta \lambda_1)^{2t} \tilde{\boldsymbol{v}}^{\top} g_0} - 1 \\
&\leq \frac{(1 - \eta \lambda_2)^{2t} \left\|g_0^{\perp}\right\|_2^2}{(1 - \eta \lambda_1)^{2t} \tilde{\boldsymbol{v}}^{\top} g_0} = \frac{(1 - \eta \lambda_2)^{2t} \left\|g_0^{\perp}\right\|_2^2}{(1 - \eta \lambda_1)^{2t} \tilde{\boldsymbol{v}}^{\top} g_0},
\end{aligned}
$$

where the first inequality follows from the fact that $\|\mathbf{G}_t\|_F^2 > \|\mathbf{G}_t\|_2^2$, the second inequality is by using Zhao et al. (2024a)[Lemma 3.3], i.e.,

$$
\|\mathbf{G}_t\|_F^2 \leq (1 - \eta \lambda_2)^{2t} \left\|g_0^{\perp}\right\|_2^2 + (1 - \eta \lambda_1)^{2t} \left\|g_0^{\|}\right\|_2^2,
$$

and the third inequality is due to Eq. equation 3. Finally, by defining the constants $c_1 \triangleq \frac{(1-\eta\lambda_2)}{(1-\eta\lambda_1)}$ and $c_2 \triangleq \frac{\left\|g_0^{\perp}\right\|_2^2}{\tilde{\boldsymbol{v}}^{\top} g_0} < 1$, we get that $\kappa(t) < c_1^{2t} \cdot c_2$, which concludes the proof.

## B.2 PROOF OF THEOREM 2

In this section, we prove Theorem 2. We upper bound Frobenius norm of $\mathbf{G}_t^j$, for any layer $j \in [L]$; in the following, for simplicity of notation, we ignore the index $j$ an use $\mathbf{G}_t$ instead. By Lemma 3, the low-rank optimization block 3 in Algorithm 3 is guaranteed to converge; we denote by $\mathsf{T}_\ell \in \mathbb{N}$ the time index $t$ at which we exit block 3 for the $\ell$th time (i.e., $\|\hat{\mathbf{G}}_{\mathsf{T}_\ell}\| \leq \varsigma_2$), for $\ell \in \mathbb{N}$. Furthermore, we recall that $\mathbf{G}_t^j \triangleq \nabla_{\mathbf{W}^j} f(\boldsymbol{\theta}_t)$; when clear from the context, we omit $j$ from $\mathbf{W}^j$, and use instead $\nabla_{\mathbf{W}^j} f(\boldsymbol{\theta}_t) = \nabla f(\mathbf{W}_t)$. Consider the SVD decomposition of the

gradient $\nabla_{\mathbf{W}^j} f(\boldsymbol{\theta}_{\mathsf{T}_i}) = \mathbf{U}_{\mathsf{T}_i} \Sigma_{\mathsf{T}_i} \mathbf{V}_{\mathsf{T}_i}^\top$. For $t \in [\mathsf{T}_i, \mathsf{T}_{i+1} - 1]$, we define the projected gradient as $\hat{\mathbf{G}}_t \triangleq \mathbf{P}_{\mathsf{T}_i}(r_{\mathsf{T}_i}) \mathbf{G}_t$, where $\mathbf{P}_{\mathsf{T}_i} = \mathbf{U}_{\mathsf{T}_i} [:, :r_{\mathsf{T}_i}]^\top$, for a given threshold $0 < \eta_{\mathrm{th}} \leq 1$, where

$$r_{\mathsf{T}_i} \triangleq \sup \left\{ r \in \mathbb{N} : \eta_{\mathrm{th}} \cdot \|\mathbf{G}_{\mathsf{T}_i}\|_F^2 - \|\mathbf{G}_{\mathsf{T}_i} - \mathbf{P}_{\mathsf{T}_i}(r)\mathbf{G}_{\mathsf{T}_i}\|_F^2 \geq 0 \right\}, \tag{4}$$

obtained by the search presented in IASS Algorithm 2, using exact truncated-SVD calculation. Note that since $\|\mathbf{G}_t - \mathbf{P}_{\mathsf{T}_i}(r)\mathbf{G}_t\|_F^2 \leq \|\mathbf{G}_t\|_F^2$ always, it must be the case that $0 < \eta_{\mathrm{th}} \leq 1$. Also, because $\mathbf{P}_{\mathsf{T}_i}(n)\mathbf{G}_t = \mathbf{G}_t$, $r_{\mathsf{T}_i}$ in equation 4 is well-defined. Next, let $h_t \triangleq f(\mathbf{W}_t) - f(\mathbf{W}_{\mathsf{T}_{i+1}})$, and $\alpha_t$ denote the learning rate. Then,

$$\begin{aligned}
h_{t+1} &= f(\mathbf{W}_{t+1}) - f(\mathbf{W}_{\mathsf{T}_{i+1}}) \\
&= f\left(\mathbf{W}_t - \alpha_t\left(\hat{\mathbf{G}}_t\right)\right) - f(\mathbf{W}_{\mathsf{T}_{i+1}}) \\
&\underset{(1)}{\leq} f(\mathbf{W}_t) - f(\mathbf{W}_{\mathsf{T}_{i+1}}) - \alpha_t \mathrm{vec}\left(\hat{\mathbf{G}}_t\right)^\top \mathrm{vec}(\nabla f(\mathbf{W}_t)) + \alpha_t^2 \frac{\beta}{2}\left\|\hat{\mathbf{G}}_t\right\|_F^2 \\
&= f(\mathbf{W}_t) - f(\mathbf{W}_{\mathsf{T}_{i+1}}) - \alpha_t \mathrm{vec}\left(\hat{\mathbf{G}}_t\right)^\top \mathrm{vec}(\mathbf{G}_t) + \alpha_t^2 \frac{\beta}{2}\left\|\hat{\mathbf{G}}_t\right\|_F^2 \\
&\underset{(2)}{\leq} f(\mathbf{W}_t) - f(\mathbf{W}_{\mathsf{T}_{i+1}}) - \alpha_t \mathrm{tr}\left((\mathbf{P}_{\mathsf{T}_i}(r_{\mathsf{T}_i})\mathbf{G}_t)^\top \mathbf{G}_t\right) + \alpha_t^2 \frac{\beta}{2}\|\mathbf{G}_t\|_F^2 \\
&\underset{(3)}{\leq} f(\mathbf{W}_t) - f(\mathbf{W}_{\mathsf{T}_{i+1}}) - \alpha_t \mathrm{tr}\left((\mathbf{G}_t^\top \mathbf{P}_{\mathsf{T}_i}(r_{\mathsf{T}_i}))\mathbf{G}_t\right) + \alpha_t^2 \frac{\beta}{2}\|\mathbf{G}_t\|_F^2 \\
&= f(\mathbf{W}_t) - f(\mathbf{W}_{\mathsf{T}_{i+1}}) - \alpha_t \|\mathbf{P}_{\mathsf{T}_i}(r_{\mathsf{T}_i})\mathbf{G}_t\|_F^2 + \alpha_t^2 \frac{\beta}{2}\|\mathbf{G}_t\|_F^2 \\
&= f(\mathbf{W}_t) - f(\mathbf{W}_{\mathsf{T}_{i+1}}) - \alpha_t \left\|\hat{\mathbf{G}}_t\right\|_F^2 + \alpha_t^2 \frac{\beta}{2}\|\mathbf{G}_t\|_F^2 \\
&\underset{(4)}{\leq} f(\mathbf{W}_t) - f(\mathbf{W}_{\mathsf{T}_{i+1}}) - \alpha_t \left\|\hat{\mathbf{G}}_t\right\|_F^2 + \frac{\beta(D\alpha_t)^2}{2} \\
&= h_t - \alpha_t \left\|\hat{\mathbf{G}}_t\right\|_F^2 + \frac{\beta(D\alpha_t)^2}{2},
\end{aligned} \tag{5}$$

where (1) follows by the assumption that $f$ is $\beta$-smooth function and the decent lemma (see, e.g. Beck (2017)[Definition 5.1]), (2) follows from the identity $\mathrm{vec}(\mathbf{AB})^\top \mathrm{vec}(\mathbf{B}) = \mathrm{tr}((\mathbf{AB})^\top \mathbf{B})$, in (3) we use the fact that since $\mathbf{P}_t$ is an orthogonal projection matrix, we have $\mathbf{P}_t = \mathbf{P}_t^\top$, and finally, (4) follows from the assumption that the norms of the gradients are bounded by $D$. Rearranging equation 5, and choosing $\alpha_t = \alpha$, for all $t \geq 0$, we readily obtain that,

$$\sum_{t=\mathsf{T}_i}^{\mathsf{T}_{i+1}-1} \left\|\hat{\mathbf{G}}_t\right\|_F^2 \leq \frac{h_{\mathsf{T}_i} - h_{\mathsf{T}_{i+1}}}{\alpha} + \frac{(\mathsf{T}_{i+1} - \mathsf{T}_i)\beta D^2 \alpha}{2}. \tag{6}$$

Thus, for $N \in \mathbb{N}$,

$$\begin{aligned}
\frac{1}{\mathsf{T}_N} \sum_{i=0}^{N-1} \sum_{t=\mathsf{T}_i}^{\mathsf{T}_{i+1}-1} \left\|\hat{\mathbf{G}}_t\right\|_F^2 &\leq \frac{1}{\mathsf{T}_N} \sum_{i=0}^{N-1} \left[\frac{h_{\mathsf{T}_i} - h_{\mathsf{T}_{i+1}}}{\alpha} + \frac{(\mathsf{T}_{i+1} - \mathsf{T}_i)\beta D^2 \alpha}{2}\right] \tag{7} \\
&= \frac{h_{\mathsf{T}_0} - h_{\mathsf{T}_N}}{\alpha \mathsf{T}_N} + \frac{(\mathsf{T}_N - \mathsf{T}_0)\beta D^2 \alpha}{2\mathsf{T}_N} \tag{8} \\
&= \frac{R}{\sqrt{\mathsf{T}_N}}, \tag{9}
\end{aligned}$$

where $R \triangleq \sqrt{4M\beta D^2}$, and in the second equality we choose $\alpha = \sqrt{\frac{4M}{\mathsf{T}_N \beta D^2}}$.[5] Now recall that by the definition of $r_{\mathsf{T}_i}$ in equation 4, we have,

$$\|\mathbf{G}_{\mathsf{T}_i} - \mathbf{P}_{\mathsf{T}_i}(r_{\mathsf{T}_i})\mathbf{G}_{\mathsf{T}_i}\|_F^2 \leq \eta_{\mathrm{th}} \|\mathbf{G}_{\mathsf{T}_i}\|_F^2. \tag{10}$$

---

[5]Note, that this choice of $\alpha$ implies the guarantees of Lemma 3, because $\alpha = \sqrt{\frac{4M}{\mathsf{T}_N \beta D^2}} \leq \sqrt{\frac{4M}{(\mathsf{T}_{i+1} - \mathsf{T}_i)\beta D^2}} \triangleq \alpha_i'$, and $\alpha_i'$ is the optimized learning rate for the $i$th block, that was chosen in the proof of Lemma 3.

Moreover, the following clearly holds for any $t \in \mathbb{N}$,

$$\|\mathbf{G}_t\|_F^2 = \|\mathbf{P}_{\mathsf{T}_i}(r_{\mathsf{T}_i})\mathbf{G}_t\|_F^2 + \|\mathbf{P}_{\mathsf{T}_i}^{\perp}(r_{\mathsf{T}_i})\mathbf{G}_t\|_F^2 \tag{11}$$

$$= \|\mathbf{P}_{\mathsf{T}_i}(r_{\mathsf{T}_i})\mathbf{G}_t\|_F^2 + \|\mathbf{G}_t - \mathbf{P}_{\mathsf{T}_i}(r_{\mathsf{T}_i})\mathbf{G}_t\|_F^2, \tag{12}$$

and thus by plugging equation 10 into equation 12, at $t = \mathsf{T}_i$, for any $i \in \mathbb{N}$, we get,

$$(1 - \eta_{\text{th}})\|\mathbf{G}_{\mathsf{T}_i}\|_F^2 \leq \|\mathbf{P}_{\mathsf{T}_i}(r_{\mathsf{T}_i})\mathbf{G}_{\mathsf{T}_i}\|_F^2. \tag{13}$$

Accordingly,

$$\left\|\mathbf{P}_{\mathsf{T}_i}^{\perp}(r_{\mathsf{T}_i})\mathbf{G}_{\mathsf{T}_i}\right\|_F^2 \leq \frac{\eta_{\text{th}}}{1 - \eta_{\text{th}}} \left\|\mathbf{P}_{\mathsf{T}_i}(r_{\mathsf{T}_i})\mathbf{G}_{\mathsf{T}_i}\right\|_F^2, \tag{14}$$

Recall from Subsection 3.1 that for the reversible layer,

$$\|\mathbf{G}_t\|_F^2 = \|(I - \alpha\mathbf{S})\mathbf{G}_{t-1}\|_F^2 \tag{15}$$

$$\leq \|(I - \alpha\mathbf{S})\|_2^2 \|\mathbf{G}_{t-1}\|_F^2 \tag{16}$$

$$= \max_i |1 - \alpha\lambda_i|^2 \|\mathbf{G}_{t-1}\|_F^2, \tag{17}$$

where $\{\lambda_i\}_i$ are the eigenvalue of $\mathbf{S}$. Thus, using the fact that $\mathbf{S}$ is positive semi-definite matrix, if the learning rate $\alpha$ is such that $\alpha \leq \frac{2}{\lambda_{\max}}$, where $\lambda_{\max}$ is the maximal eigenvalue of $\mathbf{S}$, then we get that $\max_i |1 - \alpha\lambda_i|^2 \leq 1$, and accordingly, $\|\mathbf{G}_t\|_F^2 \leq \|\mathbf{G}_{t-1}\|_F^2$. This means that the Frobenius norm of the gradient is monotonically decreasing as a function of $t$.

Now, recall that $\varsigma_{2,i}$ is any positive number such that $\varsigma_{2,i} < \sqrt{1 - \eta_{\text{th}}} \cdot \|\mathbf{G}_{\mathsf{T}_{i-1}}\|_F$. In light of equation 13, this necessarily implies that in each block $i$, we will execute (at least once) the low-rank optimization block (indeed, the condition $\|\hat{\mathbf{G}}_{\mathsf{T}_i}\|_F > \varsigma_{2,i}$ is satisfied). This, conjugated with the monotonicity property that $\|\mathbf{G}_t\|_F^2 \leq \|\mathbf{G}_{\mathsf{T}_i}\|_F^2$, for any $t \in [\mathsf{T}_i, \mathsf{T}_{i+1} - 1]$ and $i \in [N]$, imply that

$$\frac{1}{\mathsf{T}_N} \sum_{i=0}^{N-1} \sum_{t=\mathsf{T}_i}^{\mathsf{T}_{i+1}-1} \|\mathbf{G}_t\|_F^2 \leq \frac{1}{\mathsf{T}_N} \sum_{i=0}^{N-1} \sum_{t=\mathsf{T}_i}^{\mathsf{T}_{i+1}-1} \|\mathbf{G}_{\mathsf{T}_i}\|_F^2 \tag{18}$$

$$\leq \frac{1}{(1 - \eta_{\text{th}})\mathsf{T}_N} \sum_{i=1}^{N-1} \sum_{t=\mathsf{T}_i}^{\mathsf{T}_{i+1}-1} \|\mathbf{P}_{\mathsf{T}_i}(r_{\mathsf{T}_i})\mathbf{G}_t\|_F^2 \tag{19}$$

$$\leq \frac{R}{(1 - \eta_{\text{th}})\sqrt{\mathsf{T}_N}}. \tag{20}$$

Accordingly, for any $\varepsilon \geq 0$, and $\mathsf{T}_N > \frac{R^2}{(1-\eta_{\text{th}})^2\varepsilon^2}$,

$$\min_{0 \leq t \leq \mathsf{T}_N} \|\mathbf{G}_t\|_F^2 \leq \frac{1}{\mathsf{T}_N} \sum_{i=0}^{N-1} \sum_{t=\mathsf{T}_i}^{\mathsf{T}_{i+1}-1} \|\mathbf{G}_t\|_F^2 \leq \varepsilon, \tag{21}$$

and thus, there exists an iteration index $t \in [0, \mathsf{T}_N]$ for which,

$$\|\mathbf{G}_t\|_F^2 \leq \varepsilon, \tag{22}$$

which, by definition, implies that Algorithm 3 achieves an $\varepsilon$-critical point.

**Lemma 3** (Convergence of low-rank optimization block). *Consider the same setting and assumptions as in Theorem 2. Then, the time $t = \mathsf{T}_\ell \in \mathbb{N}$ at which Algorithm 3 exits block 3 for the $\ell$th time is finite, for any $\ell \in \mathbb{N}$.*

*Proof of Lemma 3.* We follow the same notations as in the proof of Theorem 2. Recall from equation 5 that the following holds true,

$$f(\mathbf{W}_{t+1}) \leq f(\mathbf{W}_t) - \alpha_t \left\|\hat{\mathbf{G}}_t\right\|_F^2 + \frac{\beta(D\alpha_t)^2}{2}, \tag{23}$$

for any $t \geq 0$. Now, we enter for the first time the low-rank optimization block of Algorithm 3 at time $\mathsf{T}_0 = t = 0$, and we next show that this block converges. Fix $\mathsf{T} \in \mathbb{N}$. Using equation 23, and following the same arguments as in equation 6–equation 9, we have,

$$\frac{1}{\mathsf{T}} \sum_{t=0}^{\mathsf{T}-1} \left\| \hat{\mathbf{G}}_t \right\|_F^2 \leq \frac{f(\mathbf{W}_0) - f(\mathbf{W}_\mathsf{T})}{\mathsf{T}\alpha} + \frac{\beta D^2 \alpha}{2} \tag{24}$$

$$\leq \frac{R}{\sqrt{\mathsf{T}}}, \tag{25}$$

where $R \triangleq \sqrt{4M\beta D^2}$, and in the second equality we choose some $\alpha \leq \sqrt{\frac{4M}{\mathsf{T}\beta D^2}}$. Accordingly, for any $\varsigma \geq 0$, and $\mathsf{T} > \frac{R^2}{\varsigma^2}$, we clearly have,

$$\min_{t \in [0, \mathsf{T}-1]} \left\| \hat{\mathbf{G}}_t \right\|_F^2 \leq \sum_{t=0}^{\mathsf{T}-1} \left\| \hat{\mathbf{G}}_t \right\|_F^2 \leq \varsigma, \tag{26}$$

namely, there exists $\mathsf{T}_1 \in [0, \mathsf{T}-1]$ such that $\left\| \hat{\mathbf{G}}_{\mathsf{T}_1} \right\|_F^2 \leq \varsigma$; note that $\mathsf{T}_1$ is the time index at which we exist the low-rank block for the first time, and the above guarantees that $\mathsf{T}_1$ finite. The same arguments above apply for any block exit time $\mathsf{T}_\ell$, $\ell \geq 2$, which concludes the proof. □

## C  ABLATION STUDY

### IMPORTANCE OF THE ADAPTIVE SUBSPACE DIMENSION

First, we note that comparing to AdaRankGrad reduces memory usage by over 25%-50% on average compared to GaLore (and LoRA), as shown in Figure 4, with respect to the layers in which the methods are applied and compared on, in fine-tuning tasks, and with 20% reduction in memory/storage per the whole final fine-tuned model, as reported in Table 4.

We emphasize that this significant benefit in reducing the memory needed in training is an exclusive consequence of the adaptivity of the (gradients) subspace dimension during training (exploiting the natural phenomenon of the decrease of the dimension in the gradients—while preserving the information ratio of any given predefined threshold).

To evaluate the contribution of adaptive subspace dimension solely to the model performance, we conduct the following experiment, in Table 5, where we fix the subspace update interval to 200 steps and study the adaptivity of the subspace dimension.

### IMPORTANCE OF THE ADAPTIVE SUBSPACE UPDATE

To assess the impact of the adaptive subspace update on model performance, we conducted the following experiment, as shown in Table 6, where we fix the rank to a constant value of `rank=4` and examine the adaptivity of the subspace updates.

Table 5: The table presents the results of an ablation experiment in which the subspace update is fixed to intervals of 200 optimization steps, while adaptivity in the subspace dimension remains enabled.

| Model | CoLA | STS-B | MRPC | RTE | SST2 | MNLI | QNLI | QQP |
|---|---|---|---|---|---|---|---|---|
| GaLore (rank=4) | 60.35 | 90.73 | 92.25 | 79.42 | 94.0 | **87.0** | 92.24 | 91.06 |
| AdaRankGrad (Initial rank=4) | **61.4** | **90.97** | **92.6** | **81.23** | **94.8** | 86.6 | **92.5** | **90.4** |
| Constant subspace time-update | | | | | | | | |
| AdaRankGrad (Initial rank=4) | 61.2 | 90.89 | 92.58 | 81.18 | 94.63 | 86.91 | 92.37 | 90.39 |

Table 6: The table presents the results of an ablation experiment in which the selected subspaces dimensions is fixed to $r = 4$, the adaptivity in the subspace updating (times) remains enabled.

| Model | CoLA | STS-B | MRPC | RTE | SST2 | MNLI | QNLI | QQP |
|---|---|---|---|---|---|---|---|---|
| GaLore (rank=4) | 60.35 | 90.73 | 92.25 | 79.42 | 94.0 | 87.0 | 92.24 | 91.06 |
| AdaRankGrad (Initial rank=4) | **61.4** | **90.97** | **92.6** | **81.23** | **94.8** | 86.6 | **92.5** | **90.4** |
| Constant rank | | | | | | | | |
| AdaRankGrad (Constant rank=4) | 61.12 | 90.81 | 92.17 | 80.13 | 94.0 | **87.12** | 92.41 | 91.29 |

## D  EFFECTIVE RANK VS $\eta_{th}$ ANALYSIS

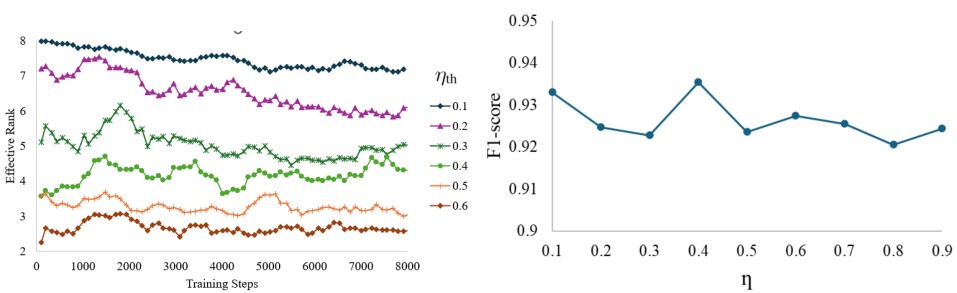

Figure 6: The left graph presents the effective rank measured for different values of $\eta_{th}$ while training AdaRankGrad on the MRPC dataset. The right graph presents the effective rank measured for different values of $\eta_{th}$ while training AdaRankGrad on the MRPC dataset.

## E  GENERALIZATION TO OTHER OPTIMIZERS

It is important to clarify that AdaRankGrad is an adaptable framework applicable to any optimization method. For illustration, AdaRankGrad's application with AdaFactor is presented in Algorithm 4, and AdaRankGrad's with Proximal SGD is presented in Algorithm 5. AdaRankGrad identifies a dominant low-dimensional subspace of the gradient, preserving relative information for each update. This allows any gradient-based optimization step to run within this subspace and, before updating weights, projects the gradient-based step back into the full space. This adaptability across optimizers is a core advantage of our approach.

## F  MEMORY SAVINGS UTILIZATION

Optimize system efficiency. Techniques such as PARIS and ELSA Kim et al. (2022) enable resource reallocation to concurrent tasks, enhancing GPU utilization in shared environments. Approaches such as FaaSwap Yu et al. (2024) support memory sharing, facilitating smaller-scale jobs alongside training, while frameworks like ZeRO Rajbhandari et al. (2019) leverage savings for parallel model validation or fine-tuning tasks. Additionally, the resource management techniques proposed in SuperNeurons Wang et al. (2018) explore how dynamic GPU memory partitioning and job scheduling can improve utilization across multiple jobs by redistributing memory savings to other pending or low-priority tasks. This allows for efficient balancing of training and inference workloads in heterogeneous environments. Similarly, the DELTA framework Tang et al. (2022) provides a dynamic combination of tensor swapping and recomputation, enabling more efficient use of memory savings during training by supporting overlapping workloads and improving overall system throughput. Integration with schedulers like Kubernetes `https://kubernetes.io/` allows real-time task prioritization and efficient resource distribution, maximizing throughput. These strategies demonstrate how training efficiency can translate into broader system-level gains.

---

**Algorithm 4** AdaRankGrad with AdaFactor step

---

**Input:** A Layer $\mathbf{W} \in \mathbb{R}^{n \times m}$ from $\boldsymbol{\theta}$, dataset $\mathcal{D}$, loss function $\mathcal{L}$, information thershold $0 < \eta_{\text{th}} < 1$, initial rank $r_{\text{init}}$, maximal rank $r_{\max}$, and small numbers $\varsigma_1, \varsigma_2 > 0$, relative step sizes $\{\rho_t\}_{t=1}^T$, second moment decay $\left\{\hat{\beta}_{2t}\right\}_{t=1}^T$ such that $\hat{\beta}_{21} = 0$, regularization constants $\epsilon_1$ and $\epsilon_2$, clipping threshold $d$.

**Initialization:** $t = 0$.
Sample batch $B \longleftarrow \{x_i, y_i\}_{i=1}^{|B|} \sim \mathcal{D}$
Compute batch gradient $\mathbf{G}_t \leftarrow \sum_{i=1}^{|B|} \frac{\partial}{\partial \mathbf{W}} \mathcal{L}(\Phi(x_i, \boldsymbol{\theta}), y_i)$       {For unsupervised $\mathcal{L}(\Phi(x_i, \boldsymbol{\theta}))$}
**while** $\|\mathbf{G}_t\|_F > \varsigma_1$ **do**

---

   **Block 1: Adaptive subspace selection**
   $\mathbf{Q}_t, r_t \leftarrow \text{IASS}(\mathbf{G}_t, r_{\text{init}}, r_{\max}, \eta_{\text{th}})$    {Approximated $r_t$-dimension subspace: $Q_t^{r_t \times n}$ projected matrix}

---

   **Block 2: Low-rank optimization**
   $\hat{\mathbf{G}}_t \leftarrow \mathbf{Q}_t^\top \mathbf{G}_t$          {Projected gradient on the approximated $r_t$-dimension subspace}
   **while** $\|\hat{\mathbf{G}}_t\|_F > \varsigma_2$ **do**

---

      **Block 3: AdaFactor update step**
      $\alpha_t = \max(\epsilon_2, \text{RMS}(\mathbf{W}_{t-1})) \rho_t$
      $\mathbf{R}_t = \hat{\beta}_{2t} \mathbf{R}_{t-1} + \left(1 - \hat{\beta}_{2t}\right) \left(\hat{\mathbf{G}}_t^2 + \epsilon_1 1_{r_t} 1_m^\top\right) 1_m$
      $\mathbf{C}_t = \hat{\beta}_{2t} \mathbf{C}_{t-1} + \left(1 - \hat{\beta}_{2t}\right) 1_{r_t}^\top \left(\hat{\mathbf{G}}_t^2 + \epsilon_1 1_{r_t} 1_m^\top\right)$
      $\hat{\mathbf{V}}_t = \mathbf{R}_t \mathbf{C}_t / 1_{r_t}^\top \mathbf{R}_t$
      $\mathbf{U}_t = \hat{\mathbf{G}}_t / \sqrt{\hat{\mathbf{V}}_t}$
      $\hat{\mathbf{U}}_t = \mathbf{U}_t / \max(1, \text{RMS}(\mathbf{U}_t)/d)$
      $\mathbf{W}_t = \mathbf{W}_{t-1} - \alpha_t \mathbf{Q}_t \hat{\mathbf{U}}_t$

---

      $t \leftarrow t + 1$
      Sample batch $B \longleftarrow \{x_i, y_i\}_{i=1}^{|B|} \sim \mathcal{D}$
      Compute batch gradient $\mathbf{G}_t \leftarrow \sum_{i=1}^{|B|} \frac{\partial}{\partial \mathbf{W}} \mathcal{L}(\Phi(x_i, \boldsymbol{\theta}), y_i)$
      $\hat{\mathbf{G}}_t \leftarrow \mathbf{Q}_t^\top \mathbf{G}_t$
   **end while**

---

**end while**     {Exit by convergence criteria could alternatively be defined by the number of epochs}
**Return** $\mathbf{W}_t$

---

# G ADDITIONAL COMPARISON TO LORA

For completeness, we repeated the experiment from Table 2 using a rank of 8, employing the hyperparameters specified in Table 3 of the original LoRA paper Hu et al. (2021) (note that Table 2 used the hyperparameters reported in Galore Table 4 Zhao et al. (2024a)). The hyperparameters used in this experiment are detailed in Table 7, and the corresponding results are presented in Table 8.

Table 7: Hyperparameter settings for different GLUE tasks using RoBERTa-Base, as used in Table 3 in Hu et al. (2021).

| Parameter / Task | MNLI | SST-2 | MRPC | CoLA | QNLI | QQP | RTE | STS-B |
|---|---|---|---|---|---|---|---|---|
| **Batch Size** | 16 | 16 | 16 | 32 | 32 | 16 | 32 | 16 |
| **Epochs** | 30 | 60 | 30 | 80 | 25 | 25 | 80 | 40 |

---

**Algorithm 5** AdaRankGrad with Proximal SGD step (ProxGen Yun et al. (2021))

---

**Input:** A Layer $\mathbf{W} \in \mathbb{R}^{n \times m}$ from $\boldsymbol{\theta}$, dataset $\mathcal{D}$, loss function $\mathcal{L}$, information thershold $0 < \eta_{\text{th}} < 1$, initial rank $r_{\text{init}}$, maximal rank $r_{\text{max}}$, and small numbers $\varsigma_1, \varsigma_2 > 0$, decay $\{\rho_t\}_{t=1}^T$ regularization constants $\epsilon_1$ and $\epsilon_2$.

**Initialization:** $t = 0$.
Sample batch $B \longleftarrow \{x_i, y_i\}_{i=1}^{|B|} \sim \mathcal{D}$
Compute batch gradient $\mathbf{G}_t \leftarrow \sum_{i=1}^{|B|} \frac{\partial}{\partial \mathbf{W}} \mathcal{L}(\Phi(x_i, \boldsymbol{\theta}), y_i)$           {For unsupervised $\mathcal{L}(\Phi(x_i, \boldsymbol{\theta}))$}
**while** $\|\mathbf{G}_t\|_F > \varsigma_1$ **do**

---

    **Block 1: Adaptive subspace selection**
    $\mathbf{Q}_t, r_t \leftarrow \text{IASS}(\mathbf{G}_t, r_{\text{init}}, r_{\text{max}}, \eta_{\text{th}})$     {Approximated $r_t$-dimension subspace: $Q_t^{r_t \times n}$ projected matrix}

---

    **Block 2: Moments subspaces transformation**
    $\mathbf{R}_t^{r_t \times r_{t-1}} \leftarrow \mathbf{Q}_t^\top \mathbf{Q}_{t-1}$ if $t \geq 1$, else $\mathbf{0}^{r_t \times r_{t-1}}$
    $\mathbf{M}_t^{r_t \times m} \leftarrow \mathbf{R}_t \mathbf{M}_{t-1}$, if $t \geq 1$, else $\mathbf{0}^{r_t \times m}$                 {$1^{st}$-order moment}

---

    **Block 3: Low-rank optimization**
    $\hat{\mathbf{G}}_t \leftarrow \mathbf{Q}_t^\top \mathbf{G}_t$                       {Projected gradient on the approximated $r_t$-dimension subspace}
    **while** $\|\hat{\mathbf{G}}_t\|_F > \varsigma_2$ **do**

---

        **Block 4: Proximal SGD update step**
        $\mathbf{M}_t \longleftarrow \rho_t \mathbf{M}_{t-1} + (1 - \rho_t) \hat{\mathbf{G}}_t$
        $\boldsymbol{\theta}_{t+1} \longleftarrow \text{prox}_{\alpha_t \lambda h}(\boldsymbol{\theta}_t - \alpha_t \mathbf{Q}_t \mathbf{M}_t)$
    **end while**

---

    **end while**           {Exit by convergence criteria could alternatively be defined by the number of epochs}
    **Return** $\mathbf{W}_t$

---

Table 8: Evaluation of RoBERTa-Base (LoRA) on the GLUE benchmark using the corresponding Hyperparameter settings as presented in Table 7. we present the accu- racy results for AdaRankGrad and LoRA.

| Method | MNLI | SST2 | MRPC | CoLA | QNLI | QQP | RTE | STS-B |
|---|---|---|---|---|---|---|---|---|
| LoRA | 87.5 | 95.1 | 89.7 | 63.4 | 93.3 | 90.8 | 86.6 | 91.5 |
| AdaRankGrad | 86.5 | **95.7** | **93.2** | **64.2** | **94.6** | 89.6 | **87.3** | **92.3** |

