# OpenReview forum: "AdaRankGrad: Adaptive Gradient Rank and Moments for Memory-Efficient LLMs Training and Fine-Tuning"
_ICLR.cc/2025/Conference — ICLR 2025 Poster_

### Official Review · Reviewer_TXWw · 2024-10-29

**Soundness:** 3
**Presentation:** 3
**Contribution:** 3
**Rating:** 8
**Confidence:** 4

**Summary:**

The paper presents a theoretical proof that the effective rank of the gradient matrix converges to 1. Building on this, it proposes dynamically reducing the gradient matrix rank as pre-training or fine-tuning progresses, thereby adaptively lowering memory consumption. Recognizing that SVD decomposition is computationally intensive, the paper introduces a novel randomized algorithm to efficiently approximate this decomposition. Table 1 highlights the advantages of the proposed method over LoRA and GaLore across several dimensions. Additionally, this approach is applicable to both pre-training and fine-tuning.

**Strengths:**

The paper presents a theoretical proof that the effective rank of the gradient matrix converges to 1. Building on this, it proposes dynamically reducing the gradient matrix rank as pre-training or fine-tuning progresses, thereby adaptively lowering memory consumption. Recognizing that SVD decomposition is computationally intensive, the paper introduces a novel randomized algorithm to efficiently approximate this decomposition. Additionally, this approach is applicable to both pre-training and fine-tuning.

**Weaknesses:**

Overall, this paper could be a significant algorithmic contribution, with the caveat for some clarifications on the theory and experiments. Given these clarifications in an author response, I would be willing to increase the score.

Feedback on specific claims and points for clarification:

1. The memory savings reported in Table 2 may be overstated. Since SSRF runs continuously, the runtime could exceed that of vanilla LoRA or pre-training alone. As a result, while less memory is allocated, it may be held for a longer duration. Memory savings should consider increased runtime since allocating 20% less memory for 20% longer results in only a 4% net saving.

2. It may be challenging to realize the claimed memory savings in practice. Figure 4 shows significant fluctuations in memory savings throughout fine-tuning or pre-training, making it difficult to allocate this saved memory to other concurrent tasks. A smoother, more predictable savings trend would simplify practical implementation.

3. The need for two parameters, $\eta$ and $\epsilon$, to define $(\eta, \epsilon)$-approximation is unclear. Would a single $\eta$ suffice?

**Suggestions that do not impact the score:**

- Consider removing the sentence on line 064, “still both of the …” since the paper does not seem to offer a solution for improving inference efficiency.

- In Algorithm 1 (SSRF), where $A$ is $n \times m$ and $\Omega$ is $k \times n$, the computation $A \Omega$ seems undefined.

- For Figure 5, should the x-axis be labeled as 11700, 23400, 35100 (for epochs 1, 2, 3)?

**Questions:**

1. What do the memory savings look like when factoring in the increased running time?
2. Can the savings be fully utilized by other jobs? If so, how?
3. Why is $\epsilon$ needed in defining the $(\eta, \epsilon)$-approximation?

---

> ### Author Response · Authors · 2024-11-20
> **(W1 +Q1) Memory and time analysis.**
>
> We would like to thank the reviewer for the thoughtful and detailed feedback. We are encouraged by the reviewer’s positive assessment that the paper has the potential to make a significant algorithmic contribution, pending some clarifications on the theory and experiments. We believe that our responses below address all of the reviewer’s questions and concerns and are organized according to the section numbering provided.
>
> Response to the reviewer’s noted weaknesses:
>
> **Memory and time analysis.**
>
> We acknowledge that the SSRF-based subspace selection in AdaRankGrad introduces some additional runtime compared to fixed-rank approaches like LoRA. However, the adaptive mechanism in AdaRankGrad is specifically designed to minimize frequent re-computation by updating the subspace only when the gradient convergence criteria are met. This approach reduces unnecessary computations.
>
> Our results demonstrate that, despite these adaptive updates, AdaRankGrad maintains overall memory efficiency without significantly increasing runtime. While continuous SSRF updates might suggest a trade-off between runtime and memory allocation, we have empirically observed that the adaptive frequency of rank updates allows AdaRankGrad to retain allocated memory for only a marginally longer duration than LoRA. We also plan to explore optimizations that could further reduce the frequency of SSRF updates, which would enhance runtime efficiency while supporting the reported memory gains.
>
> AdaRankGrad reduces memory usage by over 25%-50% on average compared to GaLore (and LoRA), as shown in Figure 4, with respect to the layers in which the methods are applied to, in fine-tuning tasks, and with 20% reduction in memory/storage, per the whole final fine-tuned model, as reported in Table 4. Below, we provide a detailed analysis and comparison of the training times for GaLoRE, AdaRankGrad, and LoRA. Our analysis shows that the total training time does not exceed that of GaLoRE and is competitive with LoRA. Specifically, we compare the total training time ratio of LoRA to AdaRankGrad, across all experiments listed in the first row of Table 2 (in the paper), the average ratio is 0.874972 (a more detailed analysis will be presented below). Therefore, accounting for the reduction in memory usage, the increase in running time (compared to LoRA) does not introduce overhead.
>
> We want to emphasize that AdaRankGrad’s memory reduction has practical implications beyond overall efficiency: it enables feasible computation on GPUs with limited memory capacity, which may otherwise be insufficient for training large models. By reducing the memory footprint, AdaRankGrad allows more users to perform fine-tuning or pre-training on consumer-grade GPUs, thereby making advanced model adaptation accessible to a broader range of hardware setups.
>
> **Time analysis.** Unlike GaLoRE, AdaRankGrad does not compute the exact singular value decomposition (SVD). Instead, it uses an efficient SVD approximation (Algorithm 1) with a complexity of $O(mnr)$, where $m$ and $n$ represent the dimensions of the layer ($m \times n$), and $r$ is the low-rank approximation ($r \ll m, n$). Specifically, in our experiments, $m, n$ are typically in the range of 1000–4000, and $r$ is set to 4, 8, or 16. This approximation significantly reduces the complexity compared to the exact SVD’s $O(\min(mn^2, m^2n))$, with only a negligible impact on error. AdaRankGrad adaptively determines the minimum rank $r \ll m, n$ required to preserve a specified information threshold from the full gradient. This is achieved using a binary search algorithm with a worst-case complexity of $O(\log(r_{max} - r_{min}))$, where $r_{max} \ll m, n$. Consequently, the complexity of this search is an order of magnitude lower than that of a single exact SVD computation, as performed in GaLoRE.In summary, the overall complexity of the estimated projection in AdaRankGrad is $O(\log(r_{max})rmn)$, which is substantially lower than GaLoRE’s $O(\min(mn^2, m^2n))$.

---

> ### Author Response · Authors · 2024-11-20
> **Continue (Memory and time analysis)**
>
> * Time comparison: We empirically measure the average time required to calculate the update of the subspace and compare it to the corresponding one in Galore (truncated SVD calculation). We do that w.r.t the “mrpc” experiment task conducted in Table 2 below. Recall that we run the fine-tuning in the three methods with the number of steps (steps for updating the parameters in the entire experiment being the same) in order to maintain fairness in relation to the accuracy obtained (the one that appears in Table 2 below). We used A100 Nvidia for this time measurement. The measurements were done on MRCP task:
>
> Table 2:The table shows the time measurements for subspace computations on the GULE MRPC dataset, comparing AdaRankGard, Galore, and LoRA methods.
> | **Subspace update role** | **Average time (ms) of subspace update - Initial Rank 8** | **Average number of subspace updates for the whole fine-tuning** | **Average overall overhead of subspace updates time** |
> |----------------------------------|----------------------------------------------------------------------|--------------------------------------------------------------------------------|------------------------------------------|
> | GaLore SVD Block                | 0.661094710                                                   | 35                                                                             | 23.13831488                               |
> | AdaRankGrad IASS (Block 1, Algorithm 3) | 0.16220796                                                   | 106.7                                                                          | 17.3075903062                             |
> | LoRA                             | None (no subspace update)                                           | None                                                                          | None                                      |
>
> From the table, we observe the following: The average time for searching the subspace (adaptive subspace updating) of AdaRankGrad (which is the average time of conducting Algorithm 2, IASS) is approximately 0.1622 milliseconds. In comparison, GaLore without the search process—relying solely on SVD decomposition with a predefined rank—has a significantly higher time cost of approximately 0.6611 milliseconds. However, the number of updates performed by AdaRankGrad is 106.7, whereas GaLore has a fixed number of updates at 35.  The effective overhead ratio can be calculated as:
>
> $\text {Overhead Ratio}=\frac{0.1622\times106.7}{0.6611\times35}\approx0.748.$ (*updated)
>
> This indicates that AdaRankGrad does not introduce substantial overhead for adaptive dimension search while offering notable advantages in terms of accuracy and memory efficiency.
>
> Table 3: The table provides a time comparison of the low-rank optimization step (rank=8 for all methods) across AdaRankGard, Galore, and LoRA.
> | low-rank optimization step | Time in ms |
>  |---------------|-------------|
> | GaLore update step (low rank adam step) | 0.63509|
>  | AdaRankGrad update step (Block 3 +4, Algorithm 3) | 0.63194|
> | Lora (with adam update step) | 1.348720934 |
>
> Please remember that in comparison to LORA, AdaRankGrad achieves better accuracy in fine-tuning tasks while using significantly less memory (see Table 4). Furthermore, unlike LORA, AdaRankGrad also supports pre-training.
> Although the update step for Lora takes longer, it does not have parallel calculation blocks like those used in Glore and AdaGradRank. As a result, the training time of Lora in this experiment was 0.91 times that of AdaGradRank.
>
> Lastly, we note that the average time of Block 2 in AdaRankGrad Moments subspaces transformation is measured with 0.023425 ms (the overhead of that block to the whole training is about 0.023425*106= 2.48305 sec which is neglected)

---

> > ### Comment · Reviewer_TXWw · 2024-11-21
> >
> > Thanks for the detailed break down of the time analysis, but I'm afraid it confused me more.
> >
> > 1) The overhead ratio is 23.14 / 17.31 which is 1.34 and is far off from 0.748 stated in the comment.
> >
> > 2) Table 2 is overall overhead, but Table 3 is per step time. I don't have a way to understand the overall running time. My understanding is that the overall running time of AdaRankGrad is 17.31 ms (from table 2) + 0.63194 ms (from table 3) * total_num_steps, but I don't know what's the total_num_steps.

---

> ### Author Response · Authors · 2024-11-20
> **(W3+Q3) (η,ϵ) definition**
>
> **(η,ϵ) definition** Our definition is intentionally general, accommodating cases where a matrix is of low rank and meets the definition with a non-zero epsilon rather than requiring epsilon to be zero. In our specific case, epsilon is zero (for the SVD decomposition, as discussed in section 2), but we chose to keep the definition more widely applicable.

---

> ### Author Response · Authors · 2024-11-20
> **Fixing Typos**
>
> We apologize for the inconvenience, and thank the reviewer for pointing out these Typos.
>
> (1) We remove the sentence mentioned.
>
> (2) The matrices' dimensions were corrected (in the revised version).
>
> (3) We re-created the plot with informative values in the x-axis (in the revised version).
>
> Lastly, we have uploaded a revised version of the paper, which includes additional runtime and ablation experiments, more detailed methodological explanations, and thorough proofreading to improve clarity.

---

> > ### Comment · Reviewer_TXWw · 2024-11-21
> >
> > (1) I still see a similar sentence in line 64: "Still, neither of these methods improves the inference efficiency of the fine-tuned model." It might be OK to have it there in the text. Just I was confused because AdaRankGrad does not seem to improve inference efficiency.
> >
> > (2) Thanks.
> >
> > (3) According to line 467, the training set is 187K and batch size is 16, so one epoch is 187K / 16 = 11.7K steps. Hence the x-axis in Figure 5 is at 1 epoch, 1.5 epoch, and 2 epoch, which is confusing because I'd expect to see either 1, 2, 3 epochs, or at 0.5, 1, 1.5, and 2 epochs. The current setting makes me wondering whether the authors trying to hide inconvenient facts at either epoch 0.5 or epoch 3. :)

---

> > > ### Author Response · Authors · 2024-11-22
> > > **Typos**
> > >
> > > **Typos**
> > >
> > > (1) You are correct; that sentence should have been removed. We have revised the text accordingly, omitting that part. Additionally, it is true that none of the methods, including ours, result in faster inference times. We apologize for any inconvenience this may have caused and appreciate you bringing it to our attention.
> > >
> > > (3) The values displayed on the X-axis correspond to the first three epochs: 1, 2, and 3. It is important to note that the actual dataset size is 93,600, not the previously mentioned 187,000. Additionally, the correct step values on the X-axis should be [5,850, 11,700, 17,550]. We mistakenly used a batch size of 32 instead of the correct size of 16 to create these plots and calculate the dataset size, which led to the discrepancy. We hope this explanation clarifies the issue, and we sincerely apologize for the error. To improve clarity, we updated the X-axis values to match the number of epochs. Thank you for bringing this to our attention.
> > >
> > > Furthermore, after training for two additional epochs, the results are as follows: our method achieved an accuracy of 88.69%, while GaLore reached 85.34%, and the LoRA method attained 80.81%.

---

> ### Author Response · Authors · 2024-11-20
> **(Q2) Memory savings utilization.**
>
> **Memory savings utilization.**  The memory savings achieved by our method can be utilized through methods such as dynamic resource allocation and job scheduling technologies, like those employed by companies such as Run.AI [1]. Several research papers have explored and suggested methods for efficient memory allocation and job scheduling that could leverage the memory savings [2,3].
>
> [1] https://www.run.ai/blog/dynamic-gpu-memory-solving-the-problem-of-inefficient-resource-allocation-in-inference-server
> [2] Wang, Linnan, et al. "Superneurons: dynamic GPU memory management for training deep neural networks." *Proceedings of the 23rd ACM SIGPLAN Symposium on Principles and Practice of Parallel Programming*. ACM, 2018, pp. 41–53. DOI: 10.1145/3178487.3178491.
> [3] Tang, Yu, et al. "DELTA: Dynamically Optimizing GPU Memory beyond Tensor Recomputation." *arXiv preprint* arXiv:2203.15980 (2022). URL: https://arxiv.org/abs/2203.15980.

---

> > ### Comment · Reviewer_TXWw · 2024-11-21
> >
> > Thanks for the references. Please include a brief discussion in the paper if space permits.
> >
> > BTW, the link in [1] does not appear to work.

---

> ### Author Response · Authors · 2024-11-22
> **Memory savings utilization discussion**
>
> **Memory savings utilization discussion**
>
> The correct link is:
>
> https://www.run.ai/blog/dynamic-gpu-memory-solving-the-problem-of-inefficient-resource-allocation-in-inference-servers
>
> Thank you for your suggestion. We have included a discussion on this topic in Appendix Section F of the revised version, as there is no remaining space in the main body of the paper.

---

> ### Author Response · Authors · 2024-11-22
> **Time analysis clarification**
>
> **Time analysis clarification**
>
> (1) We would be happy to provide clarification on this matter. As indicated in theTable 2 (in our previous comment), the average overall overhead for subspace update time is 17.307 for AdaRankGrad, which is lower compared to 23.138 for Galore, namely AdaRankGrad requires less time for subspace updates, by a factor of 0.748 (and not 1.34).
> In our previous comment, the numerator and denominator were flipped, but the result was displayed correctly.
>
> $\text {Overhead Ratio} = \frac{0.1622 \times 106.7}{0.6611 \times 35} \approx 0.748.$
>
> (2) To ensure a fair comparison in the experiments, we made sure that all methods perform the same number of optimization steps (single Adam step), differing only in how they utilize this total step "budget." Each step operates on a batch of samples of the same size and spans the same number of epochs. Following is a clarification.
>
> In the GLUE MRPC experiment:
>
> * Galore: performs 30 epochs, with each epoch consisting of approximately 230 optimization steps (calculated based on the total number of samples, 3,668, divided by the batch size, 16). This results in a total of 6,900 optimization steps, during which subspaces are updated every 200 steps, exactly as described in their paper [1].
>
> * AdaRankGrad: performs a total of 6,900 optimization steps with a batch size of 16, similar to Galore. However, unlike Galore, the number of optimization steps applied to each selected subspace adaptively varies (w.r.t convergence criteria). In practice, in this experiment, AdaRankGrad performs a total of 106 subspace updates.
>
> The calculation of the average running time per layer is expressed as:
>
> $ \text{Running Time} = (\text{Step Time} + \text{Backpropagation Time}) \times \text{Number of Batches} \times \text{Number of Epochs} + \text{Overall Overhead of Subspace Updates Time}. $
>
> Since the backpropagation time (the execution time for backpropagation on the layer parameters) is identical for both methods, and $\text{Backpropagation Time} \times \text{Number of Batches}=\text{Total optimization steps number}$, the time difference in time between the methods is determined by:
>
> $\Delta \text{Time} = (\text{AdaRankGrad-Step-Time} - \text{Galore-Step-Time})\times \text{Number of Batches} \times\text{Number of Epochs} + (\text{AdaRankGrad Subspace Overhead} - \text{Galore
> Subspace Overhead}). $
>
> Substituting the specific values from the tables 3 and 4 in our initial comment :
>
> $ \Delta \text{Time} = (0.63194-0.63509) \times 230 \times 30 + (17.3075903062-23.13831488)=-27.5657,$
>  in favor of AdaRankGrad (namely Galore preformed a bit slower).
>
>
> [1] Zhao, Jiawei, et al. "GaLore: Memory-Efficient LLM Training by Gradient Low-Rank Projection." arXiv preprint arXiv:2403.03507 (2024).

---

> > ### Comment · Reviewer_TXWw · 2024-11-25
> >
> > Thanks for the correction and detailed explanation. It's great to know that AdaRankGrad has a shorter running time than GaLore. My last question: How does it compare to plain LoRA?

---

> ### Author Response · Authors · 2024-11-27
> **Fine-tuning/Training time evaluation**
>
> We would like to thank the reviewer for suggesting this evaluation. To address your suggestion, we repeated the experiments detailed in Table 2 of the paper, to evaluate the training times of LoRA and AdaRankGrad. For this comparison, we set the rank for both methods to 8 (initial rank for AdaRankGrad) and used the same hyperparameters as those specified in the Table 2 experiments. The table below provides the fine-tuning times for each dataset and method.
> | Fine-Tuning Method | SST-2 (seconds) | STS-B (seconds) | RTE (seconds) | MRPC (seconds) | CoLA (seconds) | QNLI (seconds) |
> |---------------------|-----------------|-----------------|---------------|----------------|----------------|----------------|
> | LoRA               | 8049.06388      | 704.72446       | 318.10901     | 475.69764      | 1077.43260     | 13417.74367    |
> | AdaRankGrad        | 8720.54592      | 773.39814       | 337.28979     | 522.73626      | 1107.72272     | 13859.36416    |
>
> Although AdaRankGrad was approximately 7% slower than LoRA, it achieved a memory consumption of more than 60% less compared to LoRA.

---

> > ### Comment · Reviewer_TXWw · 2024-11-28
> >
> > Thanks for adding this comparison, which addressed my last concern. As a result, I have updated my score to 8. Thank you for your solid contribution to the research community!

---

### Official Review · Reviewer_x7pw · 2024-10-30

**Soundness:** 3
**Presentation:** 4
**Contribution:** 3
**Rating:** 6
**Confidence:** 3

**Summary:**

The paper studies an adaptive rank selection method for low-rank gradient update. Compared with previous low-rank gradient update methods like Galore, the proposed method involves an adaptively rank-reducing schedule to further reduce training memory. Experimental results and theoretical guarantees provide solid support for the claim in this paper.

**Strengths:**

The paper provides solid theoretical and numerical analysis for the design of their adaptive rank schedule. The method is effective for reducing the training memory while maintaining the training performance. The writing of the paper is clear.

**Weaknesses:**

My main concern is about the experiment:
- For the result in Table 2, the results of LoRA tuning seem far worse (2-3%) compared with the standard baselines. For example, the LoRA fine-tuning result for SST2 task should be around 95% based on the results reported in the original LORA paper. This result has also been validated in many other related works. I'm wondering if there exists any difference in the experiment setup that leads to this degradation.
- Training Efficiency: I noticed that the authors mentioned the issue of training time in the paper. However, I found no numerical results showing the difference in training time compared with the LoRA or Galore method. Even though the paper provides an approximation method to calculate the SVD. I assume this adaptive subspace selection process happened every iteration may still slow down the training time significantly. Also, I'm wondering if it's possible to avoid doing this subspace selection each iteration.

**Questions:**

Based on the previously mentioned weakness, I have the following questions:
- I also observed similar GLUE fine-tuning results in the Galore paper, but I didn't find the reason for the gap compared with the original LoRA paper. I'm wondering if the author could provide a detailed description of their LoRA implementation and experimental setup, including hyperparameters, training procedure, and evaluation method, and compare the accuracy here with the same setup as the original LoRA paper, which is a more widely used setup.
- For training efficiency, I know Galore slows down the training process due to the subspace transform process with a subspace frequency of 200. I'm worried the proposed method may further increase the training time due to the frequent update of subspace. Could you provide the concrete timing comparison between AdaRandGrad, LoRA and Galore for some of the GLUE tasks?
- For the training efficiency, Could you count the wall-clock time for block 1 and block 2 in Algorithm 3, compared with Block 3. I'm wondering about how much additional time is needed for the added parts.
- If possible, could you provide an ablation study showing how different frequencies of subspce selection affect the trade-off between the computational efficiency and performance? Especially, a comparison between the Galore and AdaGradRank with both a 200 subspace frequency is necessary to make the comparison fair.

I'm happy to further consider my score if the authors could provide a further comparison for the wall-clock time and comparison with Galore under similar subspace frequency.

---

> ### Author Response · Authors · 2024-11-20
> **(W1) Results of LoRa from previous papers and settings**
>
> We would like to thank the reviewer for the thoughtful and thorough review. We also thank the reviewer for acknowledging the paper's experimental results and theoretical guarantees, which provide solid support for the claim in this paper.
> We believe the following responses address all the reviewer’s questions and concerns, following the provided section numbering.
>
> **(W1) Results of LoRa from previous papers and settings**
> We note that the finetuning results for LoRA reported in the manuscript were borrowed from the GaLore paper [2] (Table 4). After analyzing the gap between the original LoRa paper [1] and GaLore [2], we found that the differences reside in hyperparameters. For example, Roberta-base was trained on SST-2 task with a learning rate 1E-05 for 30 epochs in [2] (Table 7 in [2]), while the same model was trained on the same task with a learning rate 5e-04 for 60 epochs in [1] (Table 9 in [1]). For completeness, we are working on running our own evaluation of LoRa on these datasets.
>
> [1] Hu, Edward J., et al. "Lora: Low-rank adaptation of large language models." arXiv preprint arXiv:2106.09685 (2021).
>
> [2] Zhao, Jiawei, et al. "Galore: Memory-efficient llm training by gradient low-rank projection." arXiv preprint arXiv:2403.03507 (2024).

---

> > ### Author Response · Authors · 2024-11-25
> >
> > For completeness, we repeated the experiment from Table 2 (in our paper) using a rank of $8$, employing the hyperparameters specified in Table 3 of the original LoRA (note that Table 2 used the hyperparameters reported in Galore Table 4). The hyperparameters used in this experiment are detailed in Table 7, and the corresponding results are presented in Table 8  (both added to the revised version appendix section G).
> >
> > Table 7: Hyperparameter settings for different GLUE tasks using RoBERTa-Base, as used in Table
> > 3 in [3].
> >
> > | Parameter / Task | MNLI | SST-2 | MRPC | CoLA | QNLI | QQP | RTE | STS-B |
> > |------------------|------|-------|------|------|------|------|------|-------|
> > | **Batch Size**   | 16   | 16    | 16   | 32   | 32   | 16   | 32   | 16    |
> > | **Epochs**       | 30   | 60    | 30   | 80   | 25   | 25   | 80   | 40    |
> >
> >
> > Table 8: Evaluation of RoBERTa-Base (LoRA) on the GLUE benchmark using the corresponding
> > Hyperparameter settings as presented in Table 7. we present the accuracy results for AdaRankGrad
> > and LoRA.
> > | Method | MNLI | SST2 | MRPC | CoLA | QNLI | QQP | RTE | STS-B |
> > |--------|------|------|------|------|------|------|------|-------|
> > | LoRA | 87.5 | 95.1 | 89.7 | 63.4 | 93.3 | 90.8 | 86.6 | 91.5 |
> > | AdaRankGrad | 86.5 | **95.7** | **93.2** | **64.2** | **94.6** | 89.6 | **87.3** | **92.3** |
> >
> >
> > [3] Hu, Edward J., et al. "LoRA: Low-Rank Adaptation of Large Language Models." *arXiv preprint* arXiv:2106.09685 (2021).

---

> ### Author Response · Authors · 2024-11-20
> ****(W2) Runtime comparison and training efficiency****
>
> Unlike GaLore, AdaRankGrad does not compute the exact singular value decomposition (SVD). Instead, it uses an efficient SVD approximation (Algorithm 1) with a complexity of $O(mnr)$, where $m$ and $n$ represent the dimensions of the layer ($m \times n$), and $r$ is the low-rank approximation ($r \ll m, n$). Specifically, in our experiments, $m, n$ are typically in the range of 1000–4000, and $r$ is set to 4, 8. This approximation significantly reduces the complexity compared to the exact SVD’s $O(\min(mn^2, m^2n))$, with only a negligible impact on error.
>
> AdaRankGrad adaptively determines the minimum rank $r \ll m, n$ required to preserve a specified information threshold from the full gradient. This is achieved using a binary search algorithm with a worst-case complexity of $O(\log(r_{max} - r_{min}))$, where $r_{max} \ll m, n$. Consequently, the complexity of this search is an order of magnitude lower than that of a single exact SVD computation, as performed in GaLoRE.In summary, the overall complexity of the estimated projection in AdaRankGrad is $\leq O(\log(r_{max})rmn)$, which is substantially lower than GaLoRE’s $O(\min(mn^2, m^2n))$.
>
> * Time comparison: We empirically measure the average time required to calculate the update of the subspace and compare it to the corresponding one in Galore (truncated SVD calculation). We do that w.r.t the “mrpc” experiment task conducted in Table 2 below. Recall that we run the fine-tuning in the three methods with the number of steps (steps for updating the parameters in the entire experiment being the same) in order to maintain fairness in relation to the accuracy obtained (the one that appears in Table 2 below). We used A100 Nvidia for this time measurement. The measurements were done on MRCP task:
>
> Table 2:The table shows the time measurements for subspace computations on the GULE MRPC dataset, comparing AdaRankGard, Galore, and LoRA methods.
> | **Subspace update role** | **Average time (ms) of subspace update - Initial Rank 8** | **Average number of subspace updates for the whole fine-tuning** | **Average overall overhead of subspace updates time** |
> |----------------------------------|----------------------------------------------------------------------|--------------------------------------------------------------------------------|------------------------------------------|
> | GaLore SVD Block                | 0.661094710                                                   | 35                                                                             | 23.13831488                               |
> | AdaRankGrad IASS (Block 1, Algorithm 3) | 0.16220796                                                   | 106.7                                                                          | 17.3075903062                             |
> | LoRA                             | None (no subspace update)                                           | None                                                                          | None                                      |
>
> From the table, we observe the following: The average time for searching the subspace (adaptive subspace updating) of AdaRankGrad (which is the average time of conducting Algorithm 2, IASS) is approximately 0.1622 milliseconds. In comparison, GaLore without the search process—relying solely on SVD decomposition with a predefined rank—has a significantly higher time cost of approximately 0.6611 milliseconds. However, the number of updates performed by AdaRankGrad is 106.7, whereas GaLore has a fixed number of updates at 35.  The effective overhead ratio can be calculated as:
>
> $\text {Overhead Ratio}=\frac{0.1622\times106.7}{0.6611\times35}\approx0.748.$ (*updated)
>
> This indicates that AdaRankGrad does not introduce substantial overhead for adaptive dimension search while offering notable advantages in terms of accuracy and memory efficiency.
>
> Table 3: The table provides a time comparison of the low-rank optimization step (rank=8 for all methods) across AdaRankGard, Galore, and LoRA.
> | low-rank optimization step | Time in ms |
>  |---------------|-------------|
> | GaLore update step (low rank adam step) | 0.63509|
>  | AdaRankGrad update step (Block 3 +4, Algorithm 3) | 0.63194|
> | Lora (with adam update step) | 1.348720934 |
>
> Please remember that in comparison to LORA, AdaRankGrad achieves better accuracy in fine-tuning tasks while using significantly less memory (see Table 4). Furthermore, unlike LORA, AdaRankGrad also supports pre-training.
> Although the update step for Lora takes longer, it does not have parallel calculation blocks like those used in Glore and AdaGradRank. As a result, the training time of Lora in this experiment was 0.91 times that of AdaGradRank.
>
> Lastly, we note that the average time of Block 2 in AdaRankGrad Moments subspaces transformation is measured with 0.023425 ms (the overhead of that block to the whole training is about 0.023425*106= 2.48305 sec which is neglected)

---

> > ### Author Response · Authors · 2024-11-20
> > **Addressing the reviewer’s query in W2**
> >
> > We would like to address the reviewer’s query regarding the possibility of avoiding subspace selection at every iteration. To clarify, this selection process (Block 1, Algorithm 3) is not performed during every iteration. Instead, it is conducted only after the model weights in the previously selected subspace have converged, as described in the inner loop of Algorithm 3 and Convergence Theorem 1. Specifically, a subspace is selected in the outer loop, and optimization is carried out within that subspace—typically using ADAM steps—until convergence is achieved. Once convergence is reached, the process repeats, ensuring that the dominant low-rank subspace is selected efficiently, without unnecessary redundancy.

---

> ### Author Response · Authors · 2024-11-20
> **(Q4) Ablation studies**
>
> **(Q4) Ablation studies**
>
> **Importance of the adaptive subspace dimension**. First, we note that comparing to  AdaRankGrad reduces memory usage by over 25%-50% on average compared to GaLore (and LoRA), as shown in Figure 4 (in the paper/revised version), with respect to the layers in which the methods are applied and compared on, in fine-tuning tasks, and with 20% reduction in memory/storage, per the whole final fine-tuned model, as reported in Table 4 (in the paper/revised version).
> We emphasize that this significant benefit in reducing the memory needed in training is an exclusive consequence of the adaptivity of the (gradients) subspace dimension during training (exploiting the natural phenomenon of the decrease of the dimension in the gradients - while preserving the information ratio of any given predefined threshold).
> To evaluate the contribution of adaptive subspace dimension solely to the model performance, we conduct the following experiment, in Table 5 below, where we fix the subspace update interval to 200 steps and study the adaptivity of the subspace dimension.
>
> Table 5: The table presents the results of an ablation experiment in which the subspace update is fixed to intervals of 200 optimization steps while adaptivity in the subspace dimension remains enabled.
>
> | Model Role | CoLA | STS-B | MRPC | RTE | SST2 | MNLI | QNLI | QQP |
> |------------|------|-------|------|-----|------|------|------|-----|
> | GaLore (rank=4) | 60.35 | 90.73 | 92.25 | 79.42 | 94.0 | **87.0** | 92.24 | 91.06 |
> | AdaRankGrad (Initial rank=4) | **61.4** | **90.97** | **92.6** | **81.23** | **94.8** | 86.6 | **92.5** | **90.4** |
> | Constant Subspace Time-Update AdaRankGrad (Initial rank=4) | 61.2 | 90.89 | 92.58 | 81.18 | 94.63 | 86.91 | 92.37 | 90.39 |
>
> **Importance of the adaptive subspace update**. To assess the impact of the adaptive subspace update on model performance, we conducted the following experiment, as shown in Table 6, where we fixed the rank to a constant value of rank=4 and examined the adaptivity of the subspace updates.
>
> Table 6: The table presents the results of an ablation experiment in which the selected subspace dimension is fixed to $r=4$, and the adaptivity in the subspace updating remains enabled.
>
> | Model Role | CoLA | STS-B | MRPC | RTE | SST2 | MNLI | QNLI | QQP |
> |------------|------|-------|------|-----|------|------|------|-----|
> | GaLore (rank=4) | 60.35 | 90.73 | 92.25 | 79.42 | 94.0 | 87.0 | 92.24 | 91.06 |
> | AdaRankGrad (Initial rank=4) | **61.4** | **90.97** | **92.6** | **81.23** | **94.8** | 86.6 | **92.5** | **90.4** |
> | Constant Rank AdaRankGrad (Constant rank=4) | 61.12 | 90.81 | 92.17 | 80.13 | 94.0 | **87.12** | 92.41 | 91.29 |
>
> The adaptive approach in our method allows optimization within the subspace to converge fully or to exit early when convergence is reached, reducing redundant steps and runtime.
>
> Please note that setting a different number of constant optimization update steps in each selected subspace would not guarantee convergence of the parameters within the subspace, as this approach does not act as a convergence criterion. However, in line with the reviewer's suggestion, we are pleased to present the following table.
>
> | **Ablation Study of AdaRankGrad on RTE GLUE** *(Adaptive Rank with Initial Rank = 4, 30 Epochs)* | **Accuracy (RTE GLUE)** |
> |----|---|
> | T = 100                                                                                            | 80.12%                  |
> | T = 150                                                                                            | 79.76%                  |
> | T = 200                                                                                            | 81.04%                  |
> | T = 250                                                                                            | 80.72%                  |
> | T = 300                                                                                            | 80.96%                  |
>
> Finally, we have uploaded a revised version of the paper with thorough proofreading, additional stylistic improvements, expanded descriptions of optimization algorithms incorporating AdaRankGrad, and new experimental results and comparisons in both the experiments section and appendix.

---

> ### Author Response · Authors · 2024-11-29
> **End of discussion period**
>
> As the discussion period is nearing its end, we would like to check if there are any remaining concerns regarding our paper. In response to the reviewers’ feedback, we have conducted a runtime evaluation (detailed in the last Table posted in response to reviewer TXWw) that demonstrates our method reduces the training time of Galore while remaining competitive with LoRA, all while significantly reducing memory usage. To strengthen our evaluation, we have also assessed our method's performance on a complex reasoning task using the GSM8K dataset. In these experiments, we applied a rank of 64 for both LoRA and Galore methods, an initial rank of 64 for AdaRankGrad, a batch size of 32, and fine-tuned over 10 epochs. The experiments were conducted on an NVIDIA A100 GPU. Our results, presented in the table below, show that AdaRankGrad is highly effective for this task, achieving improved accuracy compared to both LoRA and Galore, along with reduced memory usage. We appreciate the reviewers’ valuable suggestions and hope these results adequately address the concerns raised.
>
> | Phi-2 (2.7B)     	| Accuracy (0-shot) |
> |---------------|-------------------|
> | Base Model	| 15.16  %       	|
> | Galore    	| 52.24  %       	|
> | LoRA      	| 42.8 %         	|
> | AdaRankGrad   | **52.39** %        	|
>
> The average reduction in memory consumption of AdaRankGrad compared to Galore is 24.3%

---

> ### Comment · Reviewer_x7pw · 2024-12-02
>
> Thank you for your detailed response. The additional results and explanation address most of my concerns. I would like to improve my score to 6.

---

### Official Review · Reviewer_xMT8 · 2024-11-01

**Soundness:** 3
**Presentation:** 3
**Contribution:** 3
**Rating:** 8
**Confidence:** 4

**Summary:**

The paper introduces AdaRankGrad, a novel method aimed at optimizing memory efficiency and enhancing convergence rates and performance during the training or fine-tuning of large language models (LLMs). It addresses the issue of high memory and computational demands associated with the increasing size of model weights and optimizer states. AdaRankGrad leverages the observed phenomenon that the rank of layer gradients decreases over time, approaching rank one, to dynamically reduce the rank of gradients during Adam optimization. This is achieved through an efficient online-updating low-rank projections rule and a randomized-SVD scheme for identifying the projection matrix. The method supports full-parameter fine-tuning while significantly reducing memory usage compared to existing state-of-the-art methods.

**Strengths:**

1. The paper provides a theoretical foundation for the observation that the rank of the gradients in LLMs tends to decrease over the course of training, contributing to the understanding of gradient dynamics in deep learning.
2. In contrast to GaLore, which uses fixed time intervals for subspace updating and ignores momentum subspace alignment, the authors propose adaptive subspace selection and momentum subspace transformation, and experimentally illustrate the effectiveness of the two improvements.
3. The authors propose to use SSRF to replace the SVD used in Galore, which speeds up the search for subspaces.

**Weaknesses:**

1. According to Table 1, the main difference between this paper and Galore is the adaptive subspace dimension and adaptive subspace update. We consider that the existing experiments can illustrate the effectiveness of adaptive subspace updating but cannot illustrate the necessity of adaptive subspace dimension. Therefore, it is recommended that the authors include additional ablation studies focusing on the adaptive dimensionality to substantiate its importance.

**Questions:**

1. The method proposed in this paper requires the computation of the subspace in each iteration of the model training, how much time overhead does this operation incur compared to full parameter training and compared to Galore?

---

> ### Author Response · Authors · 2024-11-20
> **(W1) additional ablation studies**
>
> We would like to thank the reviewer for this thoughtful and detailed review. We appreciate the recognition of the article's solid theoretical foundation (showing gradients decay to a rank-1 structure over the course of training) and its significant contributions to memory efficiency through cost-effective updates of an adaptive subspace for gradients and moments projection and novel moment subspace transformation adjustment, which the reviewer found also results in competitive performance against the state-of-the-art.
> We believe the responses below address the reviewer’s questions and the noted weakness.
>
> **(W1) additional ablation studies**
>
> **Importance of the adaptive subspace dimension**. First, we note that comparing to  AdaRankGrad reduces memory usage by over 25%-50% on average compared to GaLore (and LoRA), as shown in Figure 4 (in the paper/revised version), with respect to the layers in which the methods are applied and compared on, in fine-tuning tasks, and with 20% reduction in memory/storage, per the whole final fine-tuned model, as reported in Table 4 (in the paper/revised version).
> We emphasize that this significant benefit in reducing the memory needed in training is an exclusive consequence of the adaptivity of the (gradients) subspace dimension during training (exploiting the natural phenomenon of the decrease of the dimension in the gradients - while preserving the information ratio of any given predefined threshold).
> To evaluate the contribution of adaptive subspace dimension solely to the model performance, we conduct the following experiment, in Table 5 below, where we fix the subspace update interval to 200 steps and study the adaptivity of the subspace dimension.
>
> Table 5: The table presents the results of an ablation experiment in which the subspace update is fixed to intervals of 200 optimization steps while adaptivity in the subspace dimension remains enabled.
>
> | Model Role | CoLA | STS-B | MRPC | RTE | SST2 | MNLI | QNLI | QQP |
> |------------|------|-------|------|-----|------|------|------|-----|
> | GaLore (rank=4) | 60.35 | 90.73 | 92.25 | 79.42 | 94.0 | **87.0** | 92.24 | 91.06 |
> | AdaRankGrad (Initial rank=4) | **61.4** | **90.97** | **92.6** | **81.23** | **94.8** | 86.6 | **92.5** | **90.4** |
> | Constant Subspace Time-Update AdaRankGrad (Initial rank=4) | 61.2 | 90.89 | 92.58 | 81.18 | 94.63 | 86.91 | 92.37 | 90.39 |
>
>
> **Importance of the adaptive subspace update**. To assess the impact of the adaptive subspace update on model performance, we conducted the following experiment, as shown in Table 6, where we fixed the rank to a constant value of rank=4 and examined the adaptivity of the subspace updates.
>
> Table 6: The table presents the results of an ablation experiment in which the selected subspace dimension is fixed to $r=4$, and the adaptivity in the subspace updating remains enabled.
>
> | Model Role | CoLA | STS-B | MRPC | RTE | SST2 | MNLI | QNLI | QQP |
> |------------|------|-------|------|-----|------|------|------|-----|
> | GaLore (rank=4) | 60.35 | 90.73 | 92.25 | 79.42 | 94.0 | 87.0 | 92.24 | 91.06 |
> | AdaRankGrad (Initial rank=4) | **61.4** | **90.97** | **92.6** | **81.23** | **94.8** | 86.6 | **92.5** | **90.4** |
> | Constant Rank AdaRankGrad (Constant rank=4) | 61.12 | 90.81 | 92.17 | 80.13 | 94.0 | **87.12** | 92.41 | 91.29 |
>
> In summary, the adaptive approach in our method allows optimization within the subspace to converge fully or to exit early when convergence is reached, reducing redundant steps and runtime.
> Finally, we have uploaded a revised version of the paper with thorough proofreading, additional stylistic improvements, expanded descriptions of optimization algorithms incorporating AdaRankGrad, and new experimental results and comparisons in both the experiments section and appendix.

---

> ### Author Response · Authors · 2024-11-20
> **(Q1) Iteration speed**
>
> **(W3) Iteration speed**
> Unlike GaLore, AdaRankGrad does not compute the exact singular value decomposition (SVD). Instead, it uses an efficient SVD approximation (Algorithm 1) with a complexity of $O(mnr)$, where $m$ and $n$ represent the dimensions of the layer ($m \times n$), and $r$ is the low-rank approximation ($r \ll m, n$). Specifically, in our experiments, $m, n$ are typically in the range of 1000–4000, and $r$ is set to 4, 8. This approximation significantly reduces the complexity compared to the exact SVD’s $O(\min(mn^2, m^2n))$, with only a negligible impact on error.
>
> AdaRankGrad adaptively determines the minimum rank $r \ll m, n$ required to preserve a specified information threshold from the full gradient. This is achieved using a binary search algorithm with a worst-case complexity of $O(\log(r_{max} - r_{min}))$, where $r_{max} \ll m, n$. Consequently, the complexity of this search is an order of magnitude lower than that of a single exact SVD computation, as performed in GaLoRE.In summary, the overall complexity of the estimated projection in AdaRankGrad is $\leq O(\log(r_{max})rmn)$, which is substantially lower than GaLoRE’s $O(\min(mn^2, m^2n))$.
>
> * Time comparison: We empirically measure the average time required to calculate the update of the subspace and compare it to the corresponding one in Galore (truncated SVD calculation). We do that w.r.t the “mrpc” experiment task conducted in Table 2 below. Recall that we run the fine-tuning in the three methods with the number of steps (steps for updating the parameters in the entire experiment being the same) in order to maintain fairness in relation to the accuracy obtained (the one that appears in Table 2 below). We used A100 Nvidia for this time measurement. The measurements were done on MRCP task:
>
> Table 2:The table shows the time measurements for subspace computations on the GULE MRPC dataset, comparing AdaRankGard, Galore, and LoRA methods.
> | **Subspace update role** | **Average time (ms) of subspace update - Initial Rank 8** | **Average number of subspace updates for the whole fine-tuning** | **Average overall overhead of subspace updates time** |
> |----------------------------------|----------------------------------------------------------------------|--------------------------------------------------------------------------------|------------------------------------------|
> | GaLore SVD Block                | 0.661094710                                                   | 35                                                                             | 23.13831488                               |
> | AdaRankGrad IASS (Block 1, Algorithm 3) | 0.16220796                                                   | 106.7                                                                          | 17.3075903062                             |
> | LoRA                             | None (no subspace update)                                           | None                                                                          | None                                      |
>
> From the table, we observe the following: The average time for searching the subspace (adaptive subspace updating) of AdaRankGrad (which is the average time of conducting Algorithm 2, IASS) is approximately 0.1622 milliseconds. In comparison, GaLore without the search process—relying solely on SVD decomposition with a predefined rank—has a significantly higher time cost of approximately 0.6611 milliseconds. However, the number of updates performed by AdaRankGrad is 106.7, whereas GaLore has a fixed number of updates at 35.  The effective overhead ratio can be calculated as:
>
> $\text {Overhead Ratio}=\frac{0.1622\times106.7}{0.6611\times35}\approx0.748.$ (*updated)
>
> This indicates that AdaRankGrad does not introduce substantial overhead for adaptive dimension search while offering notable advantages in terms of accuracy and memory efficiency.
>
> Table 3: The table provides a time comparison of the low-rank optimization step (rank=8 for all methods) across AdaRankGard, Galore, and LoRA.
> | low-rank optimization step | Time in ms |
>  |---------------|-------------|
> | GaLore update step (low rank adam step) | 0.63509|
>  | AdaRankGrad update step (Block 3 +4, Algorithm 3) | 0.63194|
> | Lora (with adam update step) | 1.34872093 |
>
> Please remember that in comparison to LORA, AdaRankGrad achieves better accuracy in fine-tuning tasks while using significantly less memory (see Table 4). Furthermore, unlike LORA, AdaRankGrad also supports pre-training.
> Although the update step for Lora takes longer, it does not have parallel calculation blocks like those used in Glore and AdaGradRank. As a result, the training time of Lora in this experiment was 0.91 times that of AdaGradRank.
>
> Lastly, we note that the average time of Block 2 in AdaRankGrad Moments subspaces transformation is measured with 0.02342 ms (the overhead of that block to the whole training is about 0.02342*106= 2.48305 sec which is neglected)

---

> ### Author Response · Authors · 2024-11-20
> **Revised version**
>
> We have uploaded a revised version of the paper with thorough proofreading, additional stylistic improvements, expanded descriptions of optimization algorithms incorporating AdaRankGrad, and new experimental results and comparisons in both the experiments section and appendix.

---

> > ### Comment · Reviewer_xMT8 · 2024-11-26
> >
> > Thank you for your detailed response, which adequately addresses our questions. I believe this paper has merit and should be accepted. Therefore, I will maintain the current score: Accept, Good Paper.

---

### Official Review · Reviewer_adxZ · 2024-11-02

**Soundness:** 3
**Presentation:** 3
**Contribution:** 3
**Rating:** 5
**Confidence:** 3

**Summary:**

The paper introduces a method named AdaRankGrad, aims at enhancing the memory efficiency of fine-tuning LLMs. The authors find and formally prove that the rank of estimated gradients gradually decrease to rank one. Inspired by the phenomenon, authors proposes an online updating low-rank projections rule to help reduce memory requirements while improving model performance. The paper provides a convergence analysis and demonstrates the effectiveness of AdaRankGrad through experiments on language and biological foundation models.

**Strengths:**

Strengths

1. Memory Efficiency: AdaRankGrad significantly reduces memory usage during training, which is crucial for large-scale models.

2. Theoretical Foundation: The paper provides a theoretical basis for the method, leveraging the phenomenon of decreasing gradient rank.

**Weaknesses:**

Weaknesses

1. Experiment Design: The paper's experiments, while informative, may not be entirely convincing. The use of weak benchmarks (GLUE) and the omission of key benchmarks such as MMLU (which focuses on common sense) and GSM8K (which assesses reasoning) limit the evaluation of AdaRankGrad's effectiveness on complex capabilities.

**Questions:**

Questions

1. Have there been any experiments conducted to assess the impact of this fine-tuning method on the complex capabilities of LLMs, such as MMLU and GSM8k benchmarks?

2. How do you determine the key hyperparameters in AdaRankGrad, such as the information threshold η_{th}? Because the influence of η_{th} seems quite severe to effective rank according to Appendix D.

3. If I understand correctly, the rank of the gradients may change. How do you ensure that AdaRankGrad can adapt to these changes without frequent manual parameter adjustments?

---

> ### Author Response · Authors · 2024-11-20
> **(Q2) Impact of threshold $\eta_{th}$**
>
> **(Q2) Impact of threshold $\eta_{th}$.** The information threshold $\eta_{th} $ is indeed a hyperparameter that influences the effective rank by controlling the amount of gradient information preserved during updates. Therefore, it is a desired property of this parameter that the effective rank gradually decreases with an increase in $\eta_{th}$  (as shown in Figure 6 in the Appendix). Notice, however, that as seen in Figure 7 in the appendix, changes to the information threshold $\eta_{th}$ have a minimal impact on performance (F1-score), with fluctuations of no more than 1.7 percent. A proper way to use this parameter is to select a memory budget and set $\eta_{th} $ so that a suitable rank is achieved, corresponding to the desired memory.

---

> ### Author Response · Authors · 2024-11-20
> **(Q3) Gradient dimensionality reduction**
>
> **(Q3) Gradient dimensionality reduction.**
>  AdaRankGrad incorporates an adaptive low-rank projection mechanism that dynamically adjusts the approximated-gradient-rank based on preserving-information criteria, allowing the model to respond automatically to changes in gradient rank throughout training. We have demonstrated that in gradient-based optimization, the dimensionality of layer gradients in LLM training naturally decreases, asymptotically approaching 1. Given a selected information threshold, each subspace update (outer loop in Algorithm 3) determines the effective rank of the subspace on which the gradient will be projected, preserving the relative information content of the full gradient. Thus, as the rank of the full gradient naturally reduces during training, the effective rank for the projected gradient automatically decreases as well, with no need for manual adjustment (see Algorithm 2, line $\eta_t \leftarrow \| \mathbf{A} - \mathbf{A}_{\text{app}, r_t} \|_F / \| \mathbf{A} \|_F $ ). The convergence threshold $\varsigma_2$ is set as constant, however, in Algorithm 3 can optionally be automatically adjusted based on the relative adaptive rank. For instance, if the initial rank is 8 and the rank of the selected subspace decreases to 6, the threshold is updated as $\varsigma_2 = \frac{6\varsigma_2}{8}$. This means convergence for each gradient direction is determined by its value falling below a predefined fixed threshold. As a result, the convergence threshold for the gradient norm is scaled according to the dimensionality of the gradient space, eliminating the need for manual parameter adjustment. In conclusion, no manual adjustment of parameters is required, as the adaptive parameters are automatically updated through the described mechanisms.
>
> Finally, we have uploaded a revised version of the paper with thorough proofreading, additional stylistic improvements, expanded descriptions of optimization algorithms incorporating AdaRankGrad, and new experimental results and comparisons in both the experiments section and appendix.

---

### Official Review · Reviewer_HugF · 2024-11-05

**Soundness:** 3
**Presentation:** 3
**Contribution:** 3
**Rating:** 8
**Confidence:** 3

**Summary:**

The authors propose AdaRankGrad that learns to adaptively perform low-rank gradient updates in order to reduce the memory requirements needed to train a large language model. They base their idea on a theorem that shows that the gradients rank reach to 1 over time. Their results on datasets like CoLA, STS-B, and MRPC show that AdaRankGrad is promising at significantly reducing memory requirements while retaining competitive performance compared to GaLore and LoRA.

**Strengths:**

- Reducing memory requirements is a crucial part of training LLMs
- The paper is well-written and the proofs are well designed.
- The experiments are comprehensive.

**Weaknesses:**

- The results of AdaRankGrad compared to GaLore and LoRA do not seem significant in Table. 2 How do you know if it is worth having yet another memory reduction method?
- How does this method compare to BlockLLM [1]? I see a lot of similarities between the two.
- How does the iteration speed of AdaRankGrad compare to others like LoRA, and GaLoRE. Wouldn't the method be slow given that it needs to estimate the projection matrix?
- While Adam is popular, we shouldn't always have to use it. How would AdaRankGrad generalize to other optimizers? it seems that AdaRankGrad is limited to Adam.


[1] https://arxiv.org/abs/2406.17296

**Questions:**

Could you address the weaknesses above?

---

> ### Author Response · Authors · 2024-11-20
> **(W1)  Performance gain by the method**
>
> We thank the reviewer for this thoughtful and comprehensive review! We also thank the reviewer for acknowledging that our paper is well-motivated and well-written, grounded in solid theory, and demonstrates competitive results against state-of-the-art methods. Regarding the weaknesses pointed out, we believe that our responses below thoroughly address all concerns and questions.
>
> **(W1)  Performance gain by the method**
> * As for memory reduction: By eliminating the need to predefine both the gradient rank and the fixed cycle length for subspace updates, AdaRankGrad simplifies the training process and reduces memory usage by over 25%-50% on average compared to GaLore (and LoRA), as shown in Figure 4 (in the paper/revised version), with respect to the layers which the methods are applied and compared on, in fine-tuning tasks, and with 20% reduction in memory/storage, per the whole pre-training model, as reported in Table 4 (in the paper/revised version).
> This adaptability allows AdaRankGrad to allocate memory efficiently where it’s needed during training, achieving significant gains in memory efficiency without compromising model performance and making it a valuable, scalable solution for diverse fine-tuning and training tasks. We believe these improvements will enhance the usability of low-rank gradient-based methods and pave the way for future advancements and memory efficiency of LLM training.
> * As for accuracy gain: AdaRankGrad demonstrates a consistently higher accuracy across a range of tasks, including improvements in tasks such as, RTE and MRPC, where adaptive low-rank gradient adjustment is especially effective (Table 2 in the paper/revised version). Significant gains in accuracy are also demonstrated compared to LoRA on the tabular dataset (see, Figure 5).

---

> > ### Author Response · Authors · 2024-11-20
> > **(W4) Generalization to other optimizers**
> >
> > **(W4) Generalization to other optimizers**
> > It is important to clarify that AdaRankGrad is an adaptable optimization framework applicable to any optimization method. To illustrate that, we have added examples in the appendix showing AdaRankGrad’s application with AdaFactor and Proximal SGD. AdaRankGrad identifies a dominant low-dimensional subspace of the gradient, preserving relative information for each update. This allows any gradient-based optimization step to run within this subspace and, before updating weights, projects the gradient-based step back into the full space. This adaptability across optimizers is a core advantage of our approach. Specifically,  for Adam, the (first and second) moments are also computed within the same projected space and undergo transformations across different spaces during the update process. In conclusion, the proposed optimization framework, AdaRankGrad, can be applied effectively using any optimization update rule for both fine-tuning and pre-training tasks.
> >
> > Lastly, following the reviewer's suggestions, we have uploaded a revised version of the paper, including thorough proofreading, additional stylistic improvements, expanded descriptions of optimization algorithms incorporating AdaRankGrad, and new experimental results and comparisons in the experiments section and in the appendix.
> >
> > [1] Ramesh, Amrutha Varshini, et al. "BlockLLM: Memory-Efficient Adaptation of LLMs by Selecting and Optimizing the Right Coordinate Blocks." arXiv preprint arXiv:2406.17296 (2024).

---

> ### Author Response · Authors · 2024-11-20
> **(W2) Comparison with BlockLLM**
>
> We thank the reviewer for referring us to this work. While both BlockLLM [1] and our method aim to reduce the memory requirements in training LLMs, there are significant differences between the methods, as listed below:
>
> * Algorithmically:
>
>     * The dimensionality reduction - in BlockLLM, a subset of the parameters is selected, and the model parameters are updated in this space (by e.g., ADAM) until a stopping criteria prompts trainable subspace update. In our method (AdaRankGrad), a subspace is selected via a projection until a stopping criterion prompts a trainable subspace update. Note that in AdaRankGrad, it is possible to have the same subspace as chosen in BlockLLM, however, the converse is not necessarily true.
>
>
>     * The sparsity/dimension of the trainable subspace - In BlockLLM, the sparsity is a hyperparameter. In contrast, in AdaRankGrad, the dimension of each subspace is determined by an information threshold, which is also a hyperparameter that directly relates to information loss.
>
> * Theoretically
>     * Our method (AdaGradRank) is provided with a theoretical convergence analysis. In particular, the method is motivated by an analysis of gradient rank (which shows that layers rank asymptotically goes to rank one). In contrast, in [1], theoretical analysis of BlockLLM is not provided, though, as mentioned in [1], the method is inspired by block coordinate descent, hence convergence of BlockLLM might be analyzed similarly to block coordinate descent with deterministic importance sampling.
>
>
> Following the reviewer's suggestion, we present an empirical comparison between our method and BlockLLM. Table 3 in our paper and Table 4 in [1] show the results of pretraining the model by our method on the C4 dataset. As indicated by the results, our method outperforms BlockLLM in terms of perplexity, as shown in the next table:
>
> Table 1: Perplexity for pre-trained models on the C4 dataset:
>
> | Model | 60M | 130M | 350M |
>  |---------------|-------|-------|-------|
> | BlockLLM | 34.31 | 25.36 | 19.02 |
>  | Galore | 34.88 | 25.36 | 18.95 |
> | AdaRankGrad | 34.24 | 25.22 | 18.91 |

---

> ### Author Response · Authors · 2024-11-20
> **(W3) Iteration speed**
>
> **(W3) Iteration speed**
> Unlike GaLore, AdaRankGrad does not compute the exact singular value decomposition (SVD). Instead, it uses an efficient SVD approximation (Algorithm 1) with a complexity of $O(mnr)$, where $m$ and $n$ represent the dimensions of the layer ($m \times n$), and $r$ is the low-rank approximation ($r \ll m, n$). Specifically, in our experiments, $m, n$ are typically in the range of 1000–4000, and $r$ is set to 4, 8. This approximation significantly reduces the complexity compared to the exact SVD’s $O(\min(mn^2, m^2n))$, with only a negligible impact on error.
>
> AdaRankGrad adaptively determines the minimum rank $r \ll m, n$ required to preserve a specified information threshold from the full gradient. This is achieved using a binary search algorithm with a worst-case complexity of $O(\log(r_{max} - r_{min}))$, where $r_{max} \ll m, n$. Consequently, the complexity of this search is an order of magnitude lower than that of a single exact SVD computation, as performed in GaLoRE.In summary, the overall complexity of the estimated projection in AdaRankGrad is $\leq O(\log(r_{max})rmn)$, which is substantially lower than GaLoRE’s $O(\min(mn^2, m^2n))$.
>
> * Time comparison: We empirically measure the average time required to calculate the update of the subspace and compare it to the corresponding one in Galore (truncated SVD calculation). We do that w.r.t the “mrpc” experiment task conducted in Table 2 below. Recall that we run the fine-tuning in the three methods with the number of steps (steps for updating the parameters in the entire experiment being the same) in order to maintain fairness in relation to the accuracy obtained (the one that appears in Table 2 below). We used A100 Nvidia for this time measurement. The measurements were done on MRCP task:
>
> Table 2:The table shows the time measurements for subspace computations on the GULE MRPC dataset, comparing AdaRankGard, Galore, and LoRA methods.
> | **Subspace update role** | **Average time (ms) of subspace update - Initial Rank 8** | **Average number of subspace updates for the whole fine-tuning** | **Average overall overhead of subspace updates time** |
> |----------------------------------|----------------------------------------------------------------------|--------------------------------------------------------------------------------|------------------------------------------|
> | GaLore SVD Block                | 0.661094710                                                   | 35                                                                             | 23.13831488                               |
> | AdaRankGrad IASS (Block 1, Algorithm 3) | 0.16220796                                                   | 106.7                                                                          | 17.3075903062                             |
> | LoRA                             | None (no subspace update)                                           | None                                                                          | None                                      |
>
> From the table, we observe the following: The average time for searching the subspace (adaptive subspace updating) of AdaRankGrad (which is the average time of conducting Algorithm 2, IASS) is approximately 0.1622 milliseconds. In comparison, GaLore without the search process—relying solely on SVD decomposition with a predefined rank—has a significantly higher time cost of approximately 0.6611 milliseconds. However, the number of updates performed by AdaRankGrad is 106.7, whereas GaLore has a fixed number of updates at 35.  The effective overhead ratio can be calculated as:
>
> $\text {Overhead Ratio}=\frac{0.1622\times106.7}{0.6611\times35}\approx0.748.$ (*updated)
>
> This indicates that AdaRankGrad does not introduce substantial overhead for adaptive dimension search while offering notable advantages in terms of accuracy and memory efficiency.
>
> Table 3: The table provides a time comparison of the low-rank optimization step (rank=8 for all methods) across AdaRankGard, Galore, and LoRA.
> | low-rank optimization step | Time in ms |
>  |---------------|-------------|
> | GaLore update step (low rank adam step) | 0.63509|
>  | AdaRankGrad update step (Block 3+4, Algorithm 3) | 0.63194|
> | Lora (with adam update step) | 1.348720934 |
>
> Please remember that in comparison to LORA, AdaRankGrad achieves better accuracy in fine-tuning tasks while using significantly less memory (see Table 4). Furthermore, unlike LORA, AdaRankGrad also supports pre-training.
> Although the update step for Lora takes longer, it does not have parallel calculation blocks like those used in Glore and AdaGradRank. As a result, the training time of Lora in this experiment was 0.91 times that of AdaGradRank.
>
> Lastly, we note that the average time of Block 2 in AdaRankGrad Moments subspaces transformation is measured with 0.02342 ms (the overhead of that block to the whole training is about 0.02342*106= 2.48305 sec which is neglected)

---

### Meta-Review · Area_Chair_Djat · 2024-12-13

**Metareview:**

In this paper, the authors proposed a memory-efficient low-rank gradient update algorithm for LLMs training and fine-tuning.

The reviewers raised some concerns and questions regarding the empirical evaluation of the proposed algorithm. Most of the concerns are addressed during the rebuttal.

The authors are encouraged to conduct experiments on the MMLU task as suggested by a reviewer to make the experimental results more convincing. Rather than that this paper has shown promising results, both empirically and theoretically, of the proposed algorithm. Therefore, I recommend an acceptance.

**Additional Comments On Reviewer Discussion:**

One reviewer still suggests the authors conduct experiments on the MMLU task.

---

### Decision · Program_Chairs · 2025-01-22

Accept (Poster)